# Potential of remote sensing of cirrus optical thickness by airborne spectral radiance measurements in different sideward viewing angles

Kevin Wolf[1], André Ehrlich[1], Tilman Hüneke[2], Klaus Pfeilsticker[2], Frank Werner[1,3], Martin Wirth[4], and Manfred Wendisch[1]

[1]Leipzig Institute for Meteorology, University of Leipzig, Leipzig, Germany

[2]Institute of Environmental Physics, University of Heidelberg, Heidelberg, Germany

[3]now at Joint Center for Earth Systems Technology, University of Maryland, Baltimore, MD, USA

[4]Institute of Atmospheric Physics, German Aerospace Center, Oberpfaffenhofen, Germany

*Correspondence to:* K. Wolf (kevin.wolf@uni-leipzig.de)

**Abstract.** Spectral radiance measurements collected in nadir and sideward viewing directions by two airborne passive solar remote sensing instruments, the Spectral Modular Airborne Radiation measurement sysTem (SMART) and the Differential Optical Absorption Spectrometer (mini-DOAS), are used to compare the remote sensing results of cirrus optical thickness $\tau$. The comparison is based on a sensitivity study using radiative transfer simulations (RTS) and on data obtained during three airborne field campaigns: the North Atlantic Rainfall VALidation (NARVAL) mission, the Mid-Latitude Cirrus Experiment (ML-CIRRUS) and the Aerosol, Cloud, Precipitation, and Radiation Interactions and Dynamics of Convective Cloud Systems (ACRIDICON) campaign. Radiative transfer simulations are used to quantify the sensitivity of measured upward radiance $I$ with respect to $\tau$, ice crystal effective radius $r_{\mathrm{eff}}$, viewing angle of the sensor $\theta_{\mathrm{V}}$, spectral surface albedo $\alpha$, and ice crystal shape. From the calculations it is concluded that sideward viewing measurements are generally suited better than radiances data from nadir direction to retrieve $\tau$ of optically thin cirrus, especially at wavelengths larger than $\lambda = 900$ nm. Using sideward instead of nadir-directed spectral radiance measurements significantly improves the sensitivity and accuracy to retrieve $\tau$ in particular for optically thin cirrus of $\tau \leq 2$.

The comparison of retrievals of $\tau$ based on nadir and sideward viewing radiance measurements from SMART, mini-DOAS and independent estimates of $\tau$ from an additional active remote sensing instrument, the Water Vapor Lidar Experiment in Space (WALES), show general agreement within the range of measurement uncertainties. For the selected example a mean $\tau$ of $0.54 \pm 0.2$ is derived from SMART, and $0.49 \pm 0.2$ by mini-DOAS nadir channels, while WALES obtained a mean value of $\tau = 0.32 \pm 0.02$ at 532 nm wavelength respectively. The mean of $\tau$ derived from the sideward viewing mini-DOAS channels is $0.26 \pm 0.2$. For the few simultaneous measurements, the mini-DOAS sideward channel measurements systematically underestimate (- 17.6 %) the nadir observations from SMART and mini-DOAS. The agreement between mini-DOAS sideard viewing channels and WALES is better, showing the advantage of using sideward viewing measurements for cloud remote sensing for $\tau \leq 1$. Therefore, we suggest sideward viewing measurements for retrievals of $\tau$ of thin cirrus because of the significantly enhanced capability of sideward viewing compared to nadir measurements.

# 1 Introduction

The impact of cirrus on the atmospheric radiative energy budget and the Earth's climate system is uncertain (IPCC, 2013), which is partly due to the limited knowledge about the formation and development of cirrus (Sausen et al., 2005). Until now it is not sufficiently quantified to what fraction homogeneous or heterogeneous ice nucleation contributes to the cirrus formation (Cziczo et al., 2013). As a result, the evolution of the cirrus microphysical properties during its life-cycle is insufficiently represented in climate models (IPCC, 2013). Further more, the influence of cirrus on the Earth's radiation budget is highly variable because it strongly depends on their microphysical properties such as ice crystal number, size and shape (Zhang et al., 1999; Chen et al., 2000; Wendisch et al., 2005, 2007; Yang et al., 2012). In particular, optically thin cirrus ($\tau \leq 0.03$), so called sub-visible cirrus (SVC), is difficult to observe and not well represented in General Circulation Models (Wiensz et al., 2013). Sub-visible cirrus may extend over large areas (Davis et al., 2010). Therefore, their influence on the energy budget of the Earth can probably not be neglected. Lee et al. (2009) estimated the annually and globally averaged radiative forcing of SVC with $+1\,\mathrm{W\,m^{-2}}$ (warming effect), while the local forcing might be significantly higher. Especially, the location and time where SVC occur determine their radiative effects. Whether SVC heat or cool the atmosphere depends on surface albedo $\alpha$, solar zenith angle $\theta_0$ and cirrus optical thickness $\tau$ (Fu and Liou, 1993). In general SVC and cirrus have a heating effect at the top-of-atmosphere (TOA) since the reduction of outgoing infrared radiation usually dominates the cooling effect due to reflection of solar radiation (McFarquhar et al., 2000; Comstock et al., 2002; Davis et al., 2010).

In order to quantify the microphysical and optical properties of SVC, which are needed to determine their radiative effects, more observations of this cloud type are required. As a consequence, several satellite missions and field studies were performed in the past, e.g., by Wang et al. (1996), Winker and Trepte (1998), Sassen et al. (2009), and Jensen et al. (2015) to establish a reliable data base on SVC. Airborne in-situ measurements by Lampert et al. (2009), Davis et al. (2010), Froyd et al. (2010), and Frey et al. (2011) were utilized to determine ice crystal size and shape of SVC. Optical and microphysical parameters derived from these measurements are used in radiative transfer simulation (RTS) and numerical weather prediction and climate modelling (Kärcher, 2002).

Despite these efforts, in-situ observations of SVC are still scarce and partly accidental due to the challenge of locating SVC. Lampert et al. (2009) sampled an Arctic SVC after it was detected by an airborne lidar. Airborne campaigns dedicated to visible cirrus, e.g., Contrail, volcanoe and Cirrus Experiment (CONCERT, Voigt et al. (2010)), Mid-Latitude Cirrus (ML-CIRRUS, Voigt et al. (2016)) and tropical cirrus sampled during the Airborne Tropical TRopopause EXperiment (ATTREX) are more frequent (Delanoe et al., 2013; Ehret et al., 2014; Gross et al., 2015; Jensen et al., 2015) and occasionally include observations of SVC. Further international airborne missions like the Tropical Composition, Cloud and Climate Coupling (TC4) (Toon et al., 2010) and the Cirrus Regional Study of Tropical Anvils and Cirrus Layers - Florida Area Cirrus Experiment (CRYSTAL-FACE) mission were conducted trying to fill the knowledge gap about the formation process and physical properties of tropical cirrus (Jensen et al., 2015).

While satellite observations are suited to study the global coverage of cirrus, their spatial and temporal resolution is still limited and can not resolve the high spatial variability of cirrus. As a consequence the 3 dimensional (3-D) radiative effects of different

cirrus properties, e.g., $\tau$, ice crystal size and shape, can not be studied using the coarse resolution of satellite remote sensing. Ground-based lidar and radar remote sensing can provide a high temporal resolution but are limited to a fixed location. In-situ airborne measurements can provide cirrus properties with both.

For passive remote sensing of cirrus nadir and sideward viewing observations are available. For nadir measurements $\tau$ and the effective radius $r_{\mathrm{eff}}$ of liquid water droplets can be retrieved by the bi-spectral reflectivity method after Twomey and Seton (1980) and Nakajima and King (1990). Ou et al. (1993), Rolland et al. (2000), and King et al. (2004) adapted this method for ice clouds by introducing some modifications with regard to the thermodynamic phase and crystal shape of the ice particles. Especially due to the crystal shape and low values of $\tau$ cirrus retrievals lead to additional uncertainties compared to liquid water clouds (Eichler et al., 2009; Fricke et al., 2014).

For low $\tau$, the reflected radiation is dominated by the surface reflection below the cirrus. This may introduce a bias in the retrieval of $\tau$ of up to $30\%$ when $\alpha$ is not accurately known or inhomogeneous (Fricke et al., 2014). Over dark ocean surfaces the radiance $I$ reflected by the cirrus might be weak and can be in the same order of magnitude as Rayleigh scattering in the atmosphere. In addition, inhomogeneities of cirrus lead to (3-D) radiative effects, which may cause a bias in the one-dimensional (1-D) radiative transfer simulations (Eichler et al., 2009). Incorrectly assumed ice crystal shapes also contribute to the retrieval uncertainty. Eichler et al. (2009) investigated the influence of ice crystal shape on derived $\tau$ and $r_{\mathrm{eff}}$. Evaluating a case study, they concluded that different shapes can lead to relative differences in $\tau$ of up to $70\%$. In a worst scenario, all these effects render retrievals of $\tau$ to become rather inaccurate. However, observations in sideward or limb viewing direction and improvements of retrieval techniques may overcome these limitations.

Limb measurements of SVC and cirrus were first introduced and utilized for satellite measurements by Woodbury and McCormick (1986). Since then, several applications based on this method were developed and are routinely be used, e.g. for trace gas measurements (Abrams et al., 1996; Wang et al., 1996; Clerbaux et al., 2003; Bourassa et al., 2005; Fu et al., 2007).

Many trace gas retrievals from aircraft, balloons and satellites are based on ultraviolet (UV)/ visible (VIS)/ near infrared (IR) sideward viewing measurements in combination with differential optical absorption spectroscopy (DOAS), e.g. performed by Platt and Stutz (2008). Compared to nadir observations, radiance measurements in limb or sideward viewing geometry are supposed to be more sensitive to optical thin clouds due to their observation geometry. One recent study was accomplished by Wiensz et al. (2013) who used satellite limb measurements especially for SVC investigation in the tropical tropopause layer. This data source improved SVC observations with respect to cloud climatology and microphysics.

In the present study, retrievals of $\tau$ base on simultaneous airborne nadir and sideward viewing observations of cirrus and are compared to elaborate the potential of sideward viewing measurements to derive optical parameters of SVC and optically thin cirrus. This includes a sensitivity study using RTS presented in Section 2 and measurements collected on board of the High Altitude and LOng range research aircraft (HALO) of the German Aerospace Center (DLR). With a maximum ceiling altitude of around 15 km HALO is capable to operate in and above SVC and cirrus in mid-latitudes and polar regions for in-situ measurements. The airborne observations are obtained with the Spectral Modular Airborne Radiation measurement sysTem (SMART) (Wendisch et al., 2001) and the Differential Optical Absorption Spectrometer (mini-DOAS) (Hüneke et al., 2017) both assembled on HALO. The instrumentation is introduced in Section 3. Observations from four campaigns, the Mid Latitude Cirrus

experiment (ML-CIRRUS), the Next-generation Aircraft Remote sensing for Validation Studies (NARVAL North and South), and the Aerosol, Cloud, Precipitation, and Radiation Interactions and Dynamics of Convective Cloud Systems (ACRIDICON-CHUVA) (Wendisch et al., 2016) are used to cross-calibrate the two individual instruments in terms of absolute radiance $I$ as presented in Section 4. In Section 5 an iterative retrieval of $\tau$ is introduced. Utilizing the cross-calibrations together with nadir and sideward viewing measurements of upward $I$, the retrieved results are presented and compared to reference measurements of $\tau$ to emphasize the advantages of sideward viewing observations. Section 6 concludes the study.

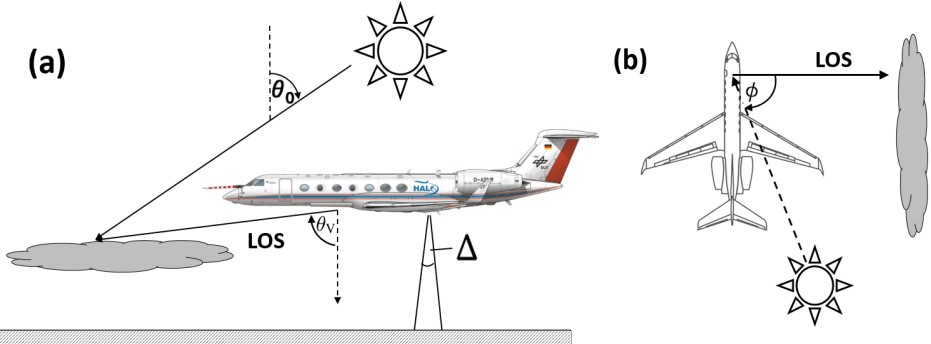

**Figure 1.** Illustration of the measurement geometry. (a) shows the side view with solar zenith angle $\theta_0$ and the viewing angle $\theta_V$. The opening angle of the nadir looking radiance sensor of SMART is indicated by $\Delta$. Top view (b) shows the definition of the relative solar azimuth angle $\phi$ between the line-of-sight (LOS) and the Sun.

## 2 Sensitivity of upward radiance measurements in nadir and sideward viewing directions

Radiative transfer simulations are performed to investigate the sensitivity of solar radiance measurements in nadir and sideward viewing geometry for SVC and thin cirrus. In this way the potential of sideward viewing versus nadir observations for cirrus cloud parameter detection is examined.

Figure 1 illustrates the measurement geometry. The solar zenith angle $\theta_0$ is the angle between zenith and the Sun. The viewing angle $\theta_V$ represents the angle of the sensor viewing direction which is measured between the Line of Sight (LOS) and the nadir direction. For a sensor measuring in nadir $\theta_V$ is $0°$ and a sensor orientation close to the horizon is around $\theta_V \approx 90°$. The relative solar azimuth angle $\phi$ represents the angle between LOS and the Sun direction. It is calculated from the difference of the azimuth angle of the Sun and the azimuth angle of the observation geometry of the optical inlets. For $\phi = 0°$ the LOS is

pointing directly in the direction of the Sun and with $\phi = 180°$ the LOS is looking away from the sun.

For the RTS a typical mid-latitude cirrus with a cloud base height of 10 km and a cloud top height of 12 km is assumed. This closely represents the cloud situation which is investigated in Section 4. Calculations are performed for $\theta_0 = 25°$, $50°$ and $75°$ representing three different scenarios. The relative solar azimuth angle is set to $\phi = 0°$, $90°$ and $180°$.

The simulations are carried out with the radiative transfer package libRadtran 2.0 (Mayer and Kylling, 2005). The Fortran 77

discrete ordinate radiative transfer solver version 2.0 (FDISORT 2) after Stamnes et al. (2000) is selected to run the simulations. The incoming extraterrestrial solar flux density given by Gueymard (2004) is applied and molecular absorption is calculated using LOWTRAN (Pierluissi and Peng, 1985). A marine aerosol profile is chosen (Shettle, 1989) and for vertical profiles of temperature, humidity, and pressure, a mid-latitude summer atmosphere profile is assumed. A spectral $\alpha$ typically for oceans is chosen according to Clark et al. (2007). To represent ice crystals, a mixture of different particle shapes is used when not other

specified. The ice crystal scattering phase function is parameterized according to Yang et al. (2013).

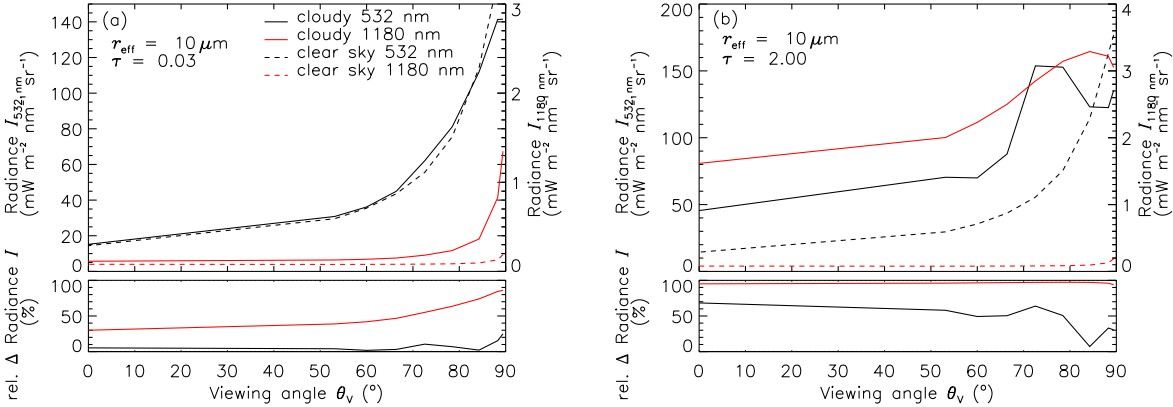

**Figure 2.** Simulated upward radiance $I_{RTS}$ at $\lambda = 532$ nm and $\lambda = 1180$ nm for cloudy (solid line) and clear sky (dashed line) case as a function of the viewing angle $\theta_V$. The left plot shows simulations for a SVC with $\tau = 0.03$ (a) and the right plot presents the simulations for a thick cirrus with $\tau = 2.0$ (b). In the corresponding lower plots the relative difference between cloud and clear sky atmosphere with respect to the cloudy atmosphere is shown.

## 2.1 Wavelength sensitivity

Using solar spectral radiation for passive remote sensing purposes, measurements at wavelengths sensitive to scattering and absorption by liquid water droplets and ice crystals are selected. Wavelengths less than $\lambda = 900$ nm are applied to retrieve $\tau$ from nadir radiance measurements. Figure 2a presents simulated upward radiances $I_{RTS}$ reflected by an optically thin cirrus

5  with $\tau = 0.03$ and $r_{eff} = 10\,\mu$m, as well as clear sky radiance as a function of the sensor viewing angle. Radiative transfer simulations for two wavelengths, $\lambda = 532$ nm and $\lambda = 1180$ nm, are carried out. To easily distinguish the different geometries, simulated $I$ in nadir geometry is denoted with $I_{RTS}^N$, while all geometries deviating from nadir are referred to sideward viewing geometry and are indicated by $I_{RTS}^V$. The sensitivity $\varepsilon_\tau$ is defined by:

$$\varepsilon_\tau = \frac{\mathrm{d}I}{\mathrm{d}\tau} \tag{1}$$

In general, $I_{RTS}^V$ increases with increasing $\theta_V$ due to the longer LOS. For a wavelength of $\lambda = 532$ nm, no difference between cloudy and clear sky conditions is discernible for all $\theta_V$, because Rayleigh scattering by molecules dominates and exceeds the scattering by thin cirrus. Therefore, at $\lambda = 532$ nm SVC with $\tau = 0.03$ which is presented in the simulations can not be detected. Conversely, for $\lambda = 1180$ nm separation between the simulations with and without cirrus at large viewing angles for

15  $\theta_V > 70°$ is present because the reflected $I_{RTS}^V$ is increased due to larger LOS. At $\lambda = 1180$ nm wavelength Rayleigh scattering is comparable weak and does not significantly contribute to the reflected radiation. In nadir direction, a detection of SVC is not possible due to low $\tau$, and the overwhelming backscattering from the ground.

For comparison, simulations of a thicker cirrus with $\tau = 2.0$ are presented in Figure 2b. Here, the influence of the Rayleigh

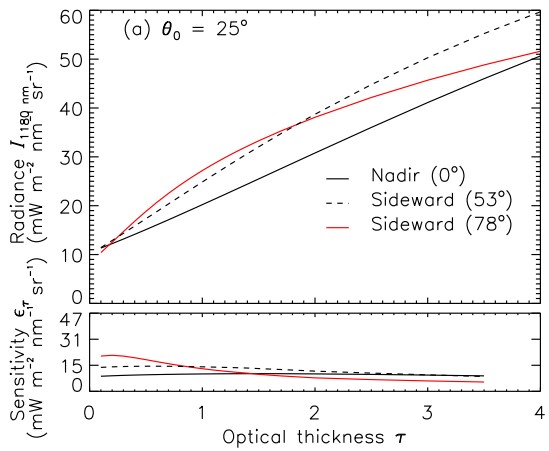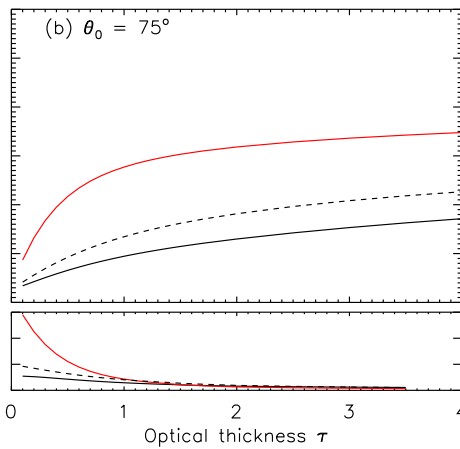

**Figure 3.** Simulated radiance $I_{RT,1180}$ for three different sensor orientations as a function of cirrus optical thickness $\tau$. Results for solar zenith angles of $\theta_0 = 25°$ (a) and $\theta_0 = 75°$ (b) are displayed. The sensitivity $\varepsilon_\tau$ is given in the lower panels.

scattering at $\lambda = 532$ nm is reduced and a distinction between cloudy and clear-sky conditions becomes possible. However, the relative difference between cloudy and clear-sky is still more pronounced at $\lambda = 1180$ nm.

The RTS suggest that sideward viewing observations at near IR wavelengths ($\lambda > 900$ nm) are more suitable for the detection of SVC and cirrus. As a result the retrieval in Section 4 is performed at 1180 nm and 1600 nm wavelength in the IR region

which are sensitive to $\tau$ and $r_{eff}$ and not disturbed by Rayleigh scattering.

## 2.2   Optical thickness and viewing angle

In general, back-scattered radiation by clouds increases with increasing $\tau$. This sensitivity (see Eq. (1)) is the basis of most retrieval algorithms of cloud optical properties. To quantify how $\varepsilon_\tau$ is effected by $\theta_V$ of the sensor, RTS are performed for a set of different $\theta_V$ ranging between $\theta_V = 0°$ (nadir) and $\theta_V = 90°$ (sideward viewing). Cirrus optical thickness is varied in the

range of $\tau = 0.03$ - 4 covering various kinds of cirrus clouds.

First simulations presented in Fig. 3 displays simulated $I_{RTS,1180}$ at $\lambda = 1180$ nm wavelength for two different $\theta_0 = 25°$ (a) and $\theta_0 = 75°$ (b) as a function of $\tau$. For each scenario, $\varepsilon_\tau$ is calculated and given in the lower panels of Fig. 3. Simulations for nadir geometry are represented by solid black lines. Results for sideward viewing sensor orientations are shown by dashed ($\theta_V = 53°$) and gray ($\theta_V = 78°$) lines. All scenarios show an increase of $I_{RTS}^V$ for increasing $\tau$, which results from enhanced

reflection.

Due to the apparent longer LOS for both $\theta_0$, sideward viewing sensor orientations yield larger $\varepsilon_\tau$ of simulated $I_{RTS}^V$ as compared to the nadir geometry for cirrus clouds with $\tau \leq 1$ which includes SVC. This indicates that sideward measurements are most suited to retrieve $\tau$ below 1 and for the detection of SVC. The almost linear increase of the nadir radiance $I_{RTS}^N$ indicates a constant $\varepsilon_\tau$ for the investigated range of $\tau$ and $\theta_0$. For $\tau \geq 1$ the sensitivity of sideward viewing observations is in the same

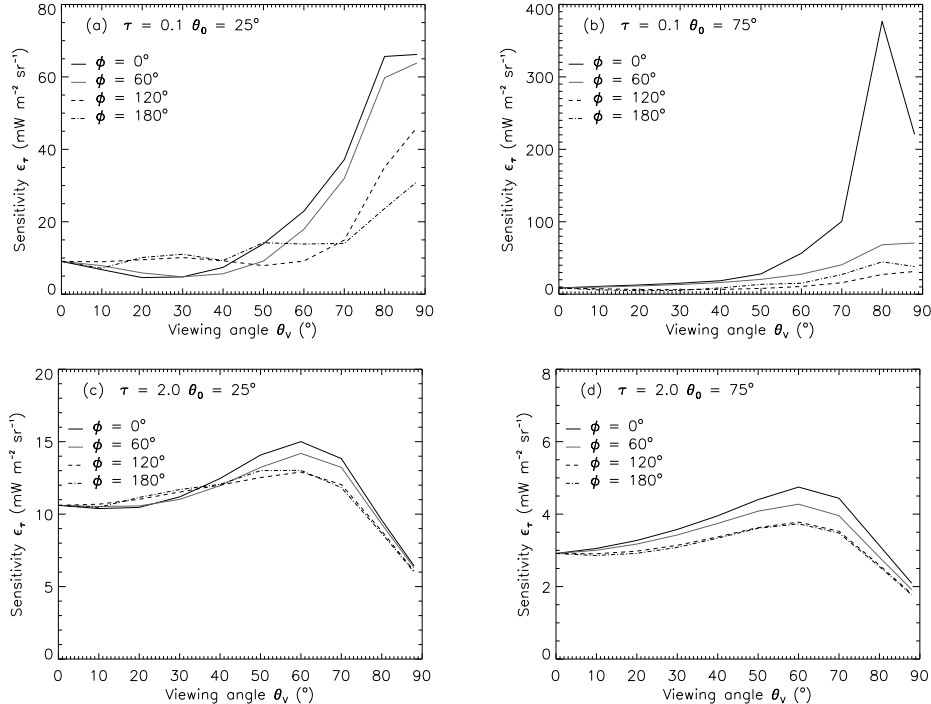

**Figure 4.** Sensitivity $\varepsilon_\tau$ at 1180 nm in units of $\mathrm{mW\,m^{-2}\,sr^{-1}}$ as a function of viewing angle $\theta_V$ and relative solar azimuth angle $\phi$ for cirrus optical thickness $\tau$ and solar zenith angle $\theta_0$. Panel (a) for $\tau = 0.1$, $\theta_0 = 25°$, Panel (b) for $\tau = 0.1$, $\theta_0 = 75°$, Panel (c) for $\tau = 2$, $\theta_0 = 25°$ in (c) and Panel (d) for $\tau = 2$, $\theta_0 = 75°$. Different scales of the plots have to be considered.

range compared to nadir measurements or slightly lower depending on the combination of $\theta_0$ and $\theta_V$.

For low $\tau$ and a high sun, the highest $\varepsilon_\tau$ is given for the sideward viewing geometry ($\theta_V = 78°$) for $\tau \leq 1$. A similar pattern emerges for low Sun ($\theta_0 = 75°$) resulting in larger $\varepsilon_\tau$ and a steep decrease for increasing $\tau$. It shows that $\varepsilon_\tau$ decreases with $\tau$ and for $\tau < 2$ drops below $\varepsilon_\tau$ of nadir measurements. The sensitivity of $I$ with respect to $\tau$ can also be interpreted in terms of

5  the uncertainty of retrieved $\tau$ related to an initial uncertainty in measured $I$. The higher $\varepsilon_\tau$ the weaker the impact of uncertainties in the measurements on the uncertainties of the retrieved $\tau$. As shown in Fig. 3b, a high $\varepsilon_\tau$ is calculated for $I_{\mathrm{RTS},1180}$ for $\tau \leq 1$ and indicates a lower measurement uncertainty. Therefore, sideward viewing observations at $\lambda = 1180$ nm allow a more accurate determination of $\tau$ compared to nadir observations for optical thin clouds with $\tau \leq 1$.

In a second step, the influence of $\phi$ is investigated on $I_{\mathrm{RTS}}^V$ in respective simulations. Figure 4 shows $\varepsilon_\tau$ for a wide range of $\theta_V$

10  between $0°$ and $90°$ and $\phi$ between $0°$ and $180°$ for two clouds with $\tau = 0.1$ and $\tau = 2$ and two different SZA of $\theta_0 = 25°$ and $\theta_0 = 75°$. The graphs represent $\varepsilon_\tau$ in units of $\mathrm{mW\,m^{-2}\,nm^{-1}\,sr^{-1}}$ for different $\phi$ as a function of $\theta_V$.

For $\tau = 0.1$ and $\theta_0 = 25°$ (Fig. 4a, $\varepsilon_\tau$ ranges between 5 and 66 $\mathrm{mW\,m^{-2}\,nm^{-1}\,sr^{-1}}$. For larger $\theta_V$ (sideward viewing observations) $\varepsilon_\tau$ increases significantly reaching the maximum for $\theta_V = 90°$ and $\phi = 0°$. Observations under these angles are better suited in comparison to other angle combinations as they enable to achieve the largest possible $\varepsilon_\tau$ and reduced relative

measurement errors which results in increased retrieval accuracy.

A similar pattern is derived for simulations assuming a lower Sun ($\theta_0 = 75°$) as shown in Fig. 4b. Compared to $\theta_0 = 25°$ the increase of $\varepsilon_\tau$ for $\theta_V = 90°$ and $\phi = 0°$ is stronger reaching values of $377\ \mathrm{mW\,m^{-2}\,nm^{-1}\,sr^{-1}}$ while for all other geometries $\varepsilon_\tau$ almost remains constant at the same magnitude reaching $80\ \mathrm{mW\,m^{-2}\,nm^{-1}\,sr^{-1}}$. Additionally, the maximum $\varepsilon_\tau$ is more

concentrated on a single combination of $\theta_V$ and $\phi$ represented by the high peak for $\phi = 0$ compared to all other $\phi$. Therefore, measurements in the range of these angles are recommended to achieve high values $\varepsilon_\tau$ for reasonable retrievals of $\tau$.

Figure 4c shows the simulated $\varepsilon_\tau$ for clouds of $\tau = 2$, $\theta_0 = 25°$ and a wide range of geometries. Compared to the optically thin cirrus, the maximum of $\varepsilon_\tau$ is reduced for optical thick cirrus not exceeding a value of $15\ \mathrm{mW\,m^{-2}\,nm^{-1}\,sr^{-1}}$ and shifted to smaller $\theta_0$. While sideward viewing measurements are predicted to become saturated for thick clouds, for low $\tau$ the optimal

$\theta_V$ is about $\theta_V = 60°$ with the largest $\varepsilon_\tau$ occurring for $\phi$ between $0°$ and $60°$. Respective simulations for $\tau = 2$, $\theta_0 = 75°$ (low Sun) are presented in Fig. 4d. Here, the maximum of $\varepsilon_\tau$ is small with $5\ \mathrm{mW\,m^{-2}\,nm^{-1}\,sr^{-1}}$ at $\theta_V$ and $\phi = 0$ compared to all other simulations varying $\tau$ and $\theta_0$.

The RTS show that the choice of the best viewing geometry (nadir or sideward viewing observations) strongly depends on $\tau$ and $\phi$. In order to probe a large range of cirrus with sufficient large retrieval sensitivity, measurements in different viewing di-

rections, at least in nadir and sideward viewing direction depending on $\tau$ and $\theta_0$ are recommended. Measurements in sideward viewing geometry strongly dependent on $\theta_V$ especially around $\theta_V = 90°$. In order to avoid spurious results by mispointing with the sensor, a careful alignment of the optical sensor and an accurate determination is required. Considering these findings, the retrieval of $\tau$ in Section 4 is performed for $\theta_V \leq 60°$ only.

## 2.3 Influence of surface albedo

The influence of $\alpha$ on the retrieval of cloud optical properties derived by passive remote sensing using the Moderate-resolution imaging spectroradiometer (MODIS) was investigated by Rolland and Liou (2001). They showed that retrievals of clouds with $\tau < 0.5$ are strongly influenced by variations in $\alpha$. Based on RTS, Fricke et al. (2014) concluded that $I^N$ measured in nadir direction strongly depends on the underlying surface reflectivity and that uncertainties in assumed $\alpha$ may cause errors of up to $50\,\%$ in the retrieval of $\tau$.

In order to quantify and compare the influence of $\alpha$ on $I$ measured in different $\theta_V$ and nadir directions, RTS are performed. To cover the natural variability of surfaces ranging from ocean surface to ice-covered regions, $\alpha$ is varied between $\alpha = 0.1$ and $\alpha = 0.9$. Figure 5 shows simulated $I^V_{\mathrm{RTS},1180}$ at $\lambda = 1180$ nm wavelength for two clouds with $\tau = 0.1$ and $\tau = 2$ and both observation geometries.

In general, the reflected $I$ increases with increasing $\alpha$. The stronger the increase, the stronger the measurements are effected

by $\alpha$. For both observation geometries, the steepest derivative,

$$\gamma = \frac{\mathrm{d}I}{\mathrm{d}\alpha},\qquad(2)$$

is obtained for the thin cirrus with $\tau = 0.1$. In general for increasing $\tau$ of thick clouds, $\alpha$ becomes less important for $I$ compared to cirrus clouds with lower $\tau$. To quantify the impact of changes in $\alpha$, the relative difference between $I_{\mathrm{RTS}}$ simulated for $\alpha = 0.1$

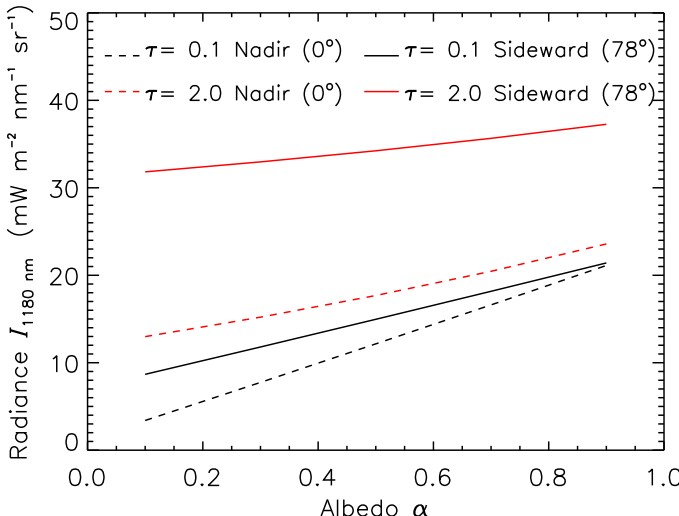

**Figure 5.** Influence of the surface albedo $\alpha$ on the measured upward radiance $I_{\mathrm{RTS},1180}^{\mathrm{V}}$ at $\lambda = 1180$ nm as a function of cirrus optical thickness $\tau$ and sensor orientation $\theta_{\mathrm{V}}$.

**Table 1.** Relative difference in $I_{\mathrm{RTS},1180\,\mathrm{nm}}$ for surface albedo $\alpha = 0.1$ and $\alpha = 0.9$ for different viewing angles $\theta_{\mathrm{V}}$ and optical thickness $\tau$.

| | cirrus optical thickness | | |
|---|---|---|---|
| viewing angle | $\tau = 0.1$ | $\tau = 0.5$ | $\tau = 2$ |
| $\theta_{\mathrm{V}} = 0°$ | 84 % | 69 % | 44 % |
| $\theta_{\mathrm{V}} = 78°$ | 58 % | 29 % | 14 % |

and $\alpha = 0.9$ is calculated for each case and presented in Table 1. Maximum differences of up to $84\%$ are noticeable in nadir geometry for clouds of $\tau = 0.1$. Optically thick clouds show lower dependencies on $\alpha$ due to the increased contribution of radiation reflected by the cirrus. Comparing nadir and sideward viewing geometries, the simulations show a smaller $\gamma$ for sideward viewing observations independent of $\alpha$. The relative difference of $I_{\mathrm{RTS}}^{\mathrm{V}}$ for $\tau = 2$ between $\alpha = 0.1$ and $\alpha = 0.9$
is reduced to $14\%$. This indicates that $I$ measured in sideward viewing geometry is less influenced by changes in $\alpha$ (e.g., Oikarinen (2002)). This difference in $I$ is most pronounced for optically thin clouds where the surface contribution to measured $I$ is relatively large. Under unknown or variable surface albedo conditions, observations in sideward viewing direction are favoured over those in nadir direction when retrieving the optical properties of thin cirrus.

## 2.4 Crystal shape sensitivity

By changing the ice crystal shape in the RTS (similar cloud as described above), the sensitivity of $I$ with respect to the ice crystal scattering phase function is investigated and compared for different viewing geometries. Ice crystals with shapes of columns, droxtals and plates are chosen and implemented in the simulations to cover the natural variability of cirrus based on

the ice crystal single scattering properties provided by Yang et al. (2013). Most cirrus are composed of a mixture of ice crystal shapes (Pruppacher and Klett, 1997). Particle shape dependent scattering effects are lower due to smoothing over different crystal shapes. Therefore, an ice crystal mixture as given by Baum et al. (2005) is included in the simulations and serves as a reference. This is denoted with the acronym 'GHM' furtheron. The simulated $I^V_{RTS,1180}$ as a function of $\theta_V$ is presented in Fig. 6.

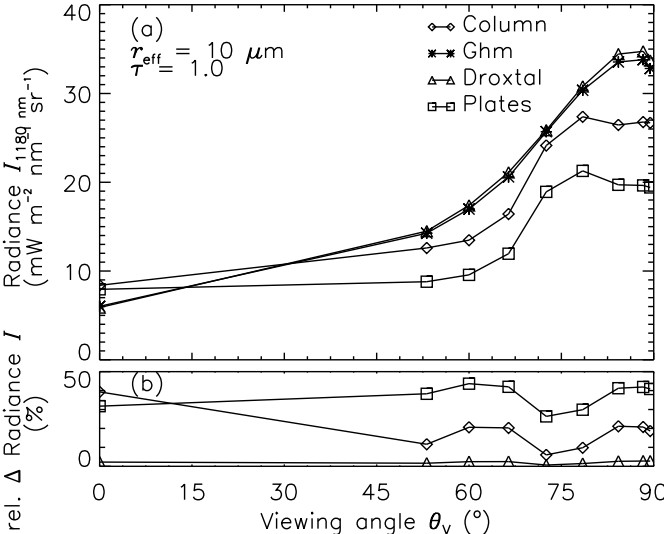

**Figure 6.** Simulated radiance $I^V_{RTS,1180}$ at $\lambda = 1180$ nm wavelength for different ice crystal shapes as a function of the viewing angle $\theta_V$ of the sensor (a). In panel (b) the relative differences of simulated radiance with respect to the reference shape 'Ghm' is presented for the three other ice crystal shapes.

The increase of $I^V_{RTS,1180}$ with increasing $\theta_V$ is significantly influenced by the ice crystal shape. In the simulated cases, droxtals and the GHM ice crystal mixture show a larger increase of $I^V_{RTS,1180}$ with increasing $\theta_V$ than columns and plates. While in nadir geometry ($\theta_V = 0°$), columns and plates have a higher $I^V_{RTS,1180}$ than droxtals and GHM, $I^V_{RTS,1180}$ measured at viewing angles $\theta_V > 50°$ is higher for droxtals and the GHM crystal mixture. The spatial distribution obtained for droxtals results from the enhanced forward and reduced sideways scattering compared to other crystal shapes.

For simulations in nadir direction the relative difference between lowest (droxtals) and highest (columns) $I^N_{RTS,1180}$ differs by up to $41.5\%$ of the absolute radiance of $6.1\,\mathrm{mW\,m^{-2}\,nm^{-1}\,sr^{-1}}$ obtained by the 'GHM' crystal mixture.

For sideward viewing observations the relative and absolute change in $I^{\mathrm{V}}_{\mathrm{RTS},1180}$ is even larger between $\theta_{\mathrm{V}} = 60°$ and $\theta_{\mathrm{V}} = 90°$. With increasing $\theta_{\mathrm{V}}$ the differences of $I^{\mathrm{V}}_{\mathrm{RTS},1180}$ increase up to a maximum of $43.5\%$ at $\theta_{\mathrm{V}} = 78°$ between droxtals and plates with respect to the absolute value of $33.8\,\mathrm{mW\,m^{-2}\,nm^{-1}\,sr^{-1}}$ for GHM.

The simulations show that the relative change in simulated $I^{\mathrm{V}}_{\mathrm{RTS},1180}$ due to ice crystal shape effects increases with $\theta_{\mathrm{V}}$. Therefore,
for cirrus of low $\tau$ the interpretation of sideward viewing observations rely even stronger on a correct assumption of ice crystal shape than nadir observations. Multiangular observations covering the angular pattern (Fig. 6), may provide sufficient information to retrieve ice crystal shape as proposed by Schäfer et al. (2013).

## 3    Airborne measurements

Simultaneous airborne measurements of $I$ in nadir and sideward viewing geometry were conducted during four campaigns
using HALO. During NARVAL shallow convection in the North Atlantic trade-wind region of the northern Atlantic (NARVAL South, December 2013) and cloud systems associated with the North Atlantic mid-latitude stormtrack (NARVAL North, January 2014) were probed (Klepp et al., 2014). During the ML-CIRRUS campaign natural and contrail cirrus in the mid-latitudes were investigated in March and April 2014 (Voigt et al., 2016). Deep convective clouds were observed during the Aerosol, Cloud, Precipitation, and Radiation Interactions and Dynamics of Convective Cloud Systems (ACRIDICON-CHHUVA) mis-
sion in September 2015 (Wendisch et al., 2016).

During these missions, a suite of different active and passive remote sensing instruments was operated on board HALO, including passive solar radiance measurements by SMART (Wendisch et al., 2016; Ehrlich et al., 2008) and the mini-DOAS (Hüneke et al., 2017). While SMART measured radiometrically calibrated radiance $I^N_S$ in nadir direction, the mini-DOAS instrument simultaneously measures in nadir and varying sideward viewing directions in UV/VIS/IR wavelength ranges. The
mini-DOAS measurements are traditionally analyzed by applying the DOAS technique. DOAS relies on an analysis of intensity ratios of two spectroscopic observations made under largely different atmospheric conditions. By exploiting ratios of $I$, DOAS measurements are inherently radiometrically calibrated in a relative but not absolute sense. Therefore no absolute radiometric calibration for $I$ for the mini-DOAS is available. In addition to the two passive sensors, active lidar measurements with the Water Vapor Lidar Experiment in Space (WALES) were performed during NARVAL and ML-CIRRUS.
In Fig. 7a the position of the apertures at the aircraft fuselage is indicated. The optical inlets of mini-DOAS and SMART for upward radiation are shown in Fig. 7b and Fig. 7c respectively.

### 3.1    The SMART instrument

Depending on the configuration, SMART measures spectral upward $F^{\uparrow}_{S,\lambda}$ and downward irradiance $F^{\downarrow}_{S,\lambda}$, as well as spectral upward radiance $I^N_S$. The system is extensively described in Wendisch et al. (2001) and Ehrlich et al. (2008). In this paper the
focus is on $I^N_S$ measurements which are available for the four HALO missions introduced above.

To cover almost the entire solar spectral range, SMART measures $I^N_S$ with two separate spectrometers, one for the VIS range from $\lambda = 300$ nm to $\lambda = 1000$ nm and a second one for sampling the IR range from $\lambda = 900$ nm to $\lambda = 2200$ nm. By merging

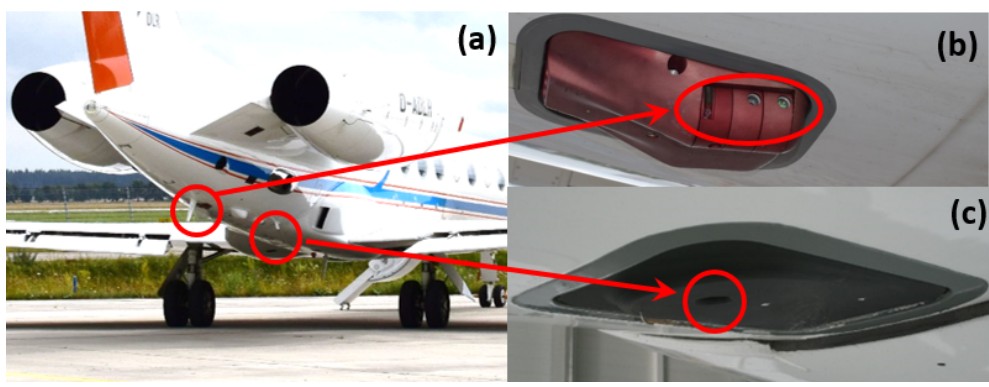

**Figure 7.** Optical inlets of mini-DOAS (b) and SMART (c) mounted at the lower aircraft fuselage.

**Table 2.** Individual sources of uncertainty and total uncertainties for the upward radiance $I_{\mathrm{S},1180}^{\mathrm{N}}$ at a wavelength of $\lambda = 1180$ nm

|  | Source of Uncertainty | $\lambda = 1180\,\mathrm{nm}$ |
|---|---|---|
| $I_{\mathrm{S},1180}^{\mathrm{N}}$ | Spectral Calibration | $< 1\%$ |
|  | Radiometric Calibration | $8.5\%$ |
|  | Signal-to-Noise-Ratio | $11.6\%$ |
|  | Transfer Calibration | $< 1.1\%$ |
|  | Total | $14.5\%$ |

the spectra, about $97\%$ of the solar spectrum is covered (Bierwirth et al., 2009, 2010). The spectral resolution defined by the full width at half maximum (FWHM) is 8 - 10 nm for the IR spectrometer and 2 - 3 nm for the VIS spectrometer.

The radiance optical inlet of SMART has an opening angle of $\Delta = 2°$ and a sampling time of 0.5 s. Considering aircraft groundspeed and the distance of 500 m between the cloud and the aircraft the resulting footprint is about 18 x 110 m for an

5 individual $I_{\mathrm{S}}^{\mathrm{N}}$ measurement. For a distance of 1000 m between sensor and cloud the footprint increases to 35 x 220 m.

Prior to each campaign SMART was radiometrically calibrated in the laboratory using certified calibration standards traceable to NIST and by secondary calibration using a travelling standard during the operation on HALO. The total measurement uncertainty of $I_{\mathrm{S}}^{\mathrm{N}}$ is about $5.4\%$ for the VIS and $14.5\%$ for the IR range which consist of individual errors due to the spectral calibration, the spectrometer noise and dark current, the radiometric calibration and the transfer calibration (Brückner et al.,

10 2014). In Table 2 the contributions of each individual source of uncertainty is given for measurements at $\lambda = 1180$ nm wavelength. The main uncertainty results from the Signal-to-Noise-Ratio (SNR) and the calibration standard, while spectral and transfer calibration errors are almost negligible. Averaging a time series of measurements will reduce the contribution of sensor noise to the signal.

## 3.2 The mini-DOAS instrument

The mini-DOAS is a passive airborne remote sensing system originally designed to retrieve vertical profiles of trace gases, aerosol and cloud particles (Hüneke et al., 2017). The analyzis is based on the DOAS technique that applies least square retrievals on the spectral shape of the observed upward radiance $I_{mD}^V$ by the mini-DOAS in sideward viewing channels

(Platt and Stutz, 2008). Spectral absorption bands of molecules and particles are measured at moderate spectral resolution (FWHM = 0.47 nm, 1.2 nm, 10 nm for the UV, VIS and IR, respectively) to quantify the absorption of solar radiation by trace gases along the light path. DOAS measurements are primarily used to infer trace gas concentrations and associated photochemistry in the atmosphere. Here, measured $I_{mD}^V$ are employed for the remote sensing of clouds.

The mini-DOAS is designed as a compact, lightweight and robust system to be operated aboard HALO. The instrument consists

of six telescopes which are connected via fiber bundles to six optical spectrometers. One set of the optical inlets is fixed in nadir configuration while the other telescopes can be tilted between $\theta_V = 0°$ and $\theta_V = 90°$. Two sets of three different spectrometers are applied to cover the UV spectral range from 310 nm to 440 nm (FWHM 0.5 nm), the VIS range from 420 nm to 650 nm (FWHM 1 nm) and the IR range from 1100 nm to 1680 nm (FWHM 10 nm). In the UV and VIS range Charged-Coupled Devices (CCD) sensors are used as detectors. The detection in the IR range is performed by Photo Diode Arrays (PDA).

The telescopes are mounted on an aperture plate at the lower side of the aircraft fuselage. The scanning telescopes have rectangular fields of view of about $0.6°$ in vertical direction and $3°$ in horizontal direction. During scanning measurements the telescopes are directed to the starboard side of the aircraft. Changes of aircraft roll angles are compensated within $0.2°$. The orientation of the nadir telescope is kept fix with respect to the aircraft major axis. Therefore no compensation of the aircraft roll angle is performed.

The evacuated spectrometer housing is immersed into an isolated water / ice tank to ensure a constant temperature and pressure of the spectrometers independent from changing outside conditions. Evacuation of the housing and temperature stabilization is necessary to guarantee a stable optical imaging, which is indispensable for DOAS applications. A spectral calibration of the spectrometers assures that wavelength shifts are less than 0.05 nm.

## 3.3 The WALES instrument

The Water Vapor Lidar Experiment in Space Demonstrator (WALES) is an airborne Differential Absorption Lidar (DIAL) with additional aerosol and cloud detection capabilities operated on the German research aircraft Falcon and HALO (Wirth et al., 2009).

For particle detection WALES has two backscatter and depolarization channels at $\lambda = 532$ nm and $\lambda = 1064$ nm wavelength and an additional high spectral resolution lidar (HSRL) channel at $\lambda = 532$ nm (Esselborn et al., 2008). The HSRL channel

allows the retrieval of the backscatter coefficient of clouds at $\lambda = 532$ nm without assumptions about the phase function of the cloud particles. Unfortunately, larger cirrus particles usually show a pronounced forward scattering peak, which may contain a significant fraction of the scattered energy. This may lead to an underestimation of $\tau$ calculated from the individual particle extinction cross sections (see e.g. (Platt, 1981)). The optical thickness data presented in this paper are corrected for the forward

scattering effect following the algorithm proposed by Eloranta (1998). To apply this correction scheme, an $r_{\mathrm{eff}}$ is assumed, which determines the width of the forward scattering peak. Best compensation of the multiple scattering decay below the cloud is found for $r_{\mathrm{eff}} = 35 \pm 5 \, \mu\mathrm{m}$ in good agreement with the climatological values proposed by Bozzo et al. (2008). The mean correction factor for the data set shown in this paper was $7\,\%$.

## 5  4  Cross-calibration

Since no radiometric calibration is available for mini-DOAS, simultaneous measurements of SMART and mini-DOAS are used to cross-calibrate the mini-DOAS with SMART. The cross-calibration relies on the radiometric calibration of SMART and allows to derive calibrated $I_{\mathrm{mD}}$ from mini-DOAS measurements. Flight sections with inhomogeneous $\alpha$ and various cloud conditions are selected to obtain a calibration valid for a wide range of different $I$. Such conditions were present during the

ML-CIRRUS flight on the 26 March 2014 including measurements over southern Germany, Belgium, United Kingdom, Ireland and the northern Atlantic Ocean westerly of Ireland. The cross-calibration is performed for the nadir and sideward viewing scanning telescopes of the mini-DOAS when aligned to the same cloud area as SMART. The results are presented for two wavelengths at $\lambda = 1180$ nm and $\lambda = 1600$ nm which are frequently used in cloud retrievals and show best discrimination potential for small $\tau$ as presented in the sensitivity study. Different FWHM of both spectrometer systems are considered by

convoluting the spectrally higher resolved measurements of the mini-DOAS with the corresponding FWHM of the SMART spectrometer (8-19 nm).

### 4.1  Nadir radiance

The nadir sensors of the mini-DOAS operate in fixed position, thus providing a large data set of simultaneous measurements with SMART. The time stamps of both instruments are corrected for temporal offsets in the data acquisition. Scatter plots of

$I_{\mathrm{S},\lambda}^{\mathrm{N}}$ and mini-DOAS raw data are shown in Fig. 8a and Fig. 8c for both wavelengths. For each data point a linear regression after Theil (1992) and Sen (1968) is performed. Using the method after Theil and Sen the influence of outliers on the regression is reduced and the linear calibration equation $I_{\mathrm{S},\lambda}^{\mathrm{N}} = a_0 \cdot N_{\mathrm{mD},\lambda}^{\mathrm{N}} + a_1$ for the mini-DOAS radiances are determined. $I_{\mathrm{S},\lambda}^{\mathrm{N}}$ is the radiance measured by SMART, $N_{\mathrm{mD},\lambda}^{\mathrm{N}}$ the raw signal of mini-DOAS and $a_0$ and $a_1$ the calibration coefficients. The linear regressions are indicated by the gray lines in Fig. 8a and Fig. 8c. For the ML-CIRRUS flight on 26 March 2014 the nadir ge-

ometry calibration coefficients are determined as $a_0 = 0.31 \, \mathrm{mW}\,\mathrm{m}^{-2}\,\mathrm{sr}^{-1}$ and $a_1 = 0.55 \, \mathrm{mW}\,\mathrm{m}^{-2}\,\mathrm{sr}^{-1}$ for $\lambda = 1180$ nm with an uncertainty of $\pm 0.24 \, \mathrm{mW}\,\mathrm{m}^{-2}\,\mathrm{sr}^{-1}$. Similar calibrations are performed for flights during the NARVAL and ACRIDICON-CHUVA campaigns. All calibration coefficients are summarized in Table 3. The coefficients depend on various environmental condition where the temperature dependence of the mini-DOAS spectrometers is the most influencing parameter.

The uncertainty is mostly related to differences of the FOV and the related difference in the observed scene and possible mi-

nor mismatches of the nadir orientation of both sensors. This means that both sensors do not always observe the exact same cloud area. For the $\lambda = 1600$ nm wavelength, $a_0$ is higher compared to $\lambda = 1180$ nm in all analyzed flights indicating the different spectral sensitivities of both sensors with SMART in comparison with mini-DOAS being relatively more sensitive at

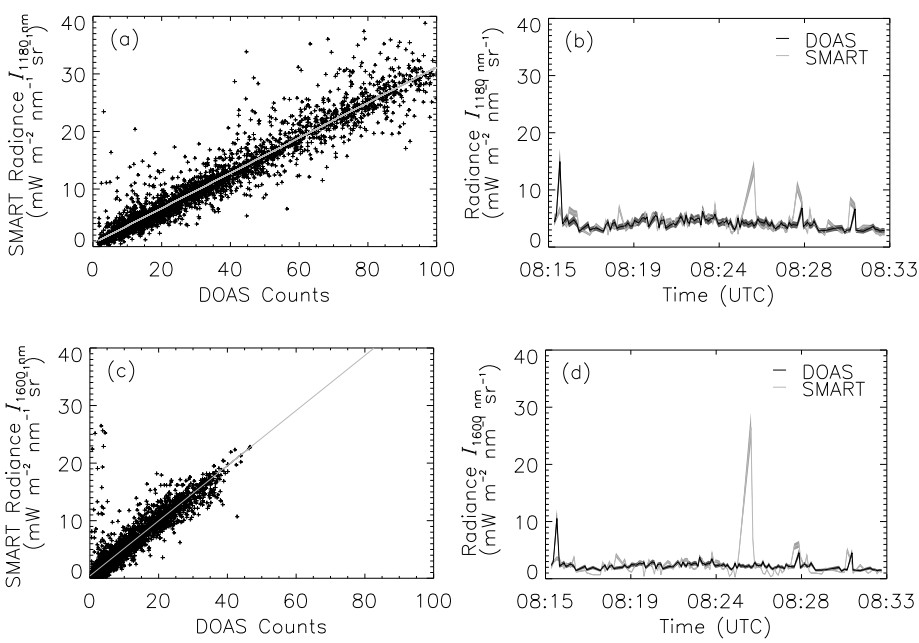

**Figure 8.** Panel (a) and (c) show comparisons of SMART radiance $I_S^N$ and mini-DOAS raw signal for nadir channels at $\lambda = 1180$ nm and $\lambda = 1600$ nm wavelength. Panel (b) and (d) show time series of measured SMART radiance $I_{S,\lambda}^N$ and calibrated mini-DOAS radiance $I_{mD,\lambda}^N$ for the ML-CIRRUS flight on 26 March 2014. The shaded areas indicate the measurement uncertainties.

$\lambda = 1600$ nm than at $\lambda = 1180$ nm wavelength.

The derived cross-calibrations of mini-DOAS are applied to all mini-DOAS measurements. A measurement example of a time series of calibrated mini-DOAS radiances $I_{mD,\lambda}^N$ is shown in Fig. 8b and 8d for a 18 minute flight section measured on the 26 March 2014.

5  The radiance time series for $\lambda = 1180$ nm of both sensors agree within the SMART error range for most data points, except for some radiance peaks. These differences likely result from the different FOV of both instruments and the presence of patches of low cumulus with high reflectivity. A similar result is obtained for $\lambda = 1600$ nm. The differences of the mean radiance between both instruments for the time period presented in Fig. 8 is $0.75\,\mathrm{mW\,m^{-2}\,nm^{-1}\,sr^{-1}}$ at $\lambda = 1180$ nm and $0.5\,\mathrm{mW\,m^{-2}\,nm^{-1}\,sr^{-1}}$ at $\lambda = 1600$ nm which results in relative differences of $5.4\,\%$ at $\lambda = 1180$ nm and $1.9\,\%$ at $\lambda = 1600$ nm

10  compared to the SMART absolute values.

### 4.2 Sideward viewing radiance

The scanning telescopes of the mini-DOAS typically run in a sequential mode scanning different $\theta_V$. During selected flight segments the scanning sequences are configured to include nadir measurements. Due to this sequential mode less measurements from the sideward viewing channels are available for cross-calibration with SMART because only measurements in nadir sen-

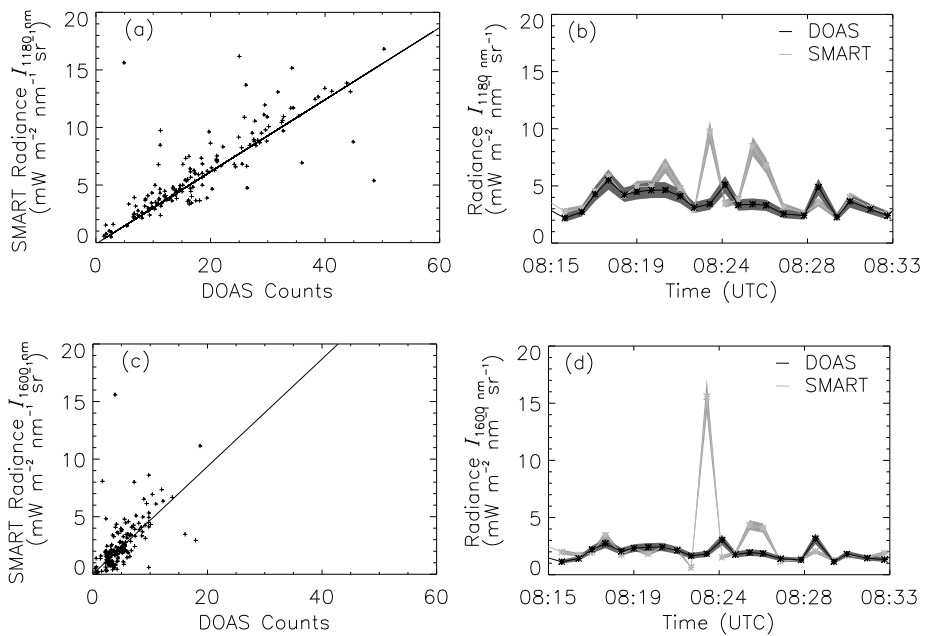

**Figure 9.** Panel (a) and (c) show a comparison of SMART radiance $I_{S,\lambda}^{N}$ and mini-DOAS raw signal $N_{mD,\lambda}^{V}$ for the scanning channels at $\lambda = 1180$ nm and $\lambda = 1600$ nm wavelength. Panel (b) and (d) show time series of measured SMART radiance $I_{S,\lambda}^{N}$ and calibrated mini-DOAS radiance $I_{mD,\lambda}^{V}$ for the ML-CIRRUS flight on 26 March 2014. The shaded areas indicate the measurement errors.

sor orientation are applicable for the cross-calibration. To ensure a statistically sufficient number of samples, the entire flight of 26 March 2014 is analyzed applying the same methods used for the calibration of the nadir channels. Figures 9a and 9c show the cross-calibration of SMART radiances $I_{S,\lambda}^{N}$ and mini-DOAS raw data $N_{mD,\lambda}^{V}$ and the linear fit (gray line) used for calibration. For the IR scanning channels the calibration coefficients are determined as $a_0 = 0.31 \, \mathrm{mW \, m^{-2} \, sr^{-1}}$ with no offset

5    $a_1$ for $\lambda = 1180$ nm and an uncertainty of $\pm 0.2 \, \mathrm{mW \, m^{-2} \, sr^{-1}}$. Similar to the nadir channels, the calibration coefficients for the the sideward viewing channel at $\lambda = 1600$ nm wavelength with $a_0 = 0.47$ are higher compared to the $\lambda = 1180$ nm wavelength. The calibration of the sideward viewing channels is repeated for the NARVAL flights while for all ACRIDICON-CHUVA flights no nadir observations of the sideward viewing channels are available. Table 3 provides a summary of all calibration coefficients derived for the sideward viewing channels.

10    Similar to Fig. 8b and Fig. 8d, Figures 9b and 9d show time series of SMART radiance $I_{S,\lambda}^{N}$ and calibrated mini-DOAS nadir observations of $I_{mD,\lambda}^{V}$ with the sideward viewing channels for a 18 minutes flight segment of the ML-Cirrus on 26 March 2014. In general, the radiance pattern observed by SMART is represented by the calibrated mini-DOAS radiance. However, individual data points differ due to differences in FOV resulting in mean differences of $0.78 \, \mathrm{mW \, m^{-2} \, nm^{-1} \, sr^{-1}}$ at 1180 nm and $0.38 \, \mathrm{mW \, m^{-2} \, nm^{-1} \, sr^{-1}}$ at $\lambda = 1600$ nm which results in relative differences of $3.7\%$ at $\lambda = 1180$ nm and $2.4\%$

**Table 3.** Calibration coefficients $a_0$ and $a_1$ in units of $\mathrm{mW\,m^{-2}\,nm^{-1}\,sr^{-1}}$ for mini-DOAS nadir and scanning channel radiance obtained for NARVAL (19 December 2013), ML-CIRRUS (26 March 2014) and ACRIDICON-CHUVA (9, 12 and 23 September 2014).

| | 1180 nm | | | | 1600 nm | | | |
| | Nadir | | sideward viewing | | Nadir | | sideward viewing | |
| | | | $(\mathrm{mW\,m^{-2}\,nm^{-1}\,sr^{-1}})$ | | | | | |
| | $a_0$ | $a_1$ | $a_0$ | $a_1$ | $a_0$ | $a_1$ | $a_0$ | $a_1$ |
|---|---|---|---|---|---|---|---|---|
| NARVAL (19.12.) | 0.26 | 5.40 | 0.23 | 0.90 | 0.28 | 1.32 | 0.26 | 0.10 |
| ML-CIRRUS (26.03.) | 0.31 | 0.55 | 0.31 | 0.00 | 0.43 | 0.25 | 0.47 | 0.02 |
| ACRIDICON-CHUVA (09.09) | 0.24 | 5.28 | | | 0.37 | 2.80 | | |
| ACRIDICON-CHUVA (12.09.) | 0.34 | 0.94 | | | 0.51 | 0.77 | | |
| ACRIDICON-CHUVA (23.09.) | 0.31 | 3.43 | | | 0.40 | 0.59 | | |

at $\lambda = 1600$ nm compared to the SMART absolute values. This ranges below the uncertainty range of SMART.

### 4.3 Temporal stability of cross-calibration

The mini-DOAS instrument is not explicitly designed to maintain a stable radiometric calibration but more for a stable wavelength calibration. For DOAS measurements absolute values of $I$ are not needed as only relative intensities are used. More important is the wavelength accuracy to determine absorption and emission bands of gasses precisely. As a result the radiometric calibration of the mini-DOAS can change from campaign to campaign and even between several flights. Therefore, cross-calibration coefficients for different campaigns and flights are derived to consider these changes of radiometric calibration and the optical setup, for example when changing the optical fibers. Using different calibration factors for the mini-DOAS instrument as inferred for the different campaigns, Fig. 10 shows a comparison of measured $I$ at $\lambda = 1180$ nm wavelength from a four minutes long flight segment over the Amazon region on 12 September 2014. The comparison clearly indicates that the measurements of $I$ of both sensors are not systematically biased and agree within the errors of each sensor except when differences at small spatial scales appear resulting from the different FOV.

The deviation of the different calibrations is below $2.9\,\mathrm{mW\,m^{-2}\,nm^{-1}\,sr^{-1}}$ which is inside the measurement uncertainties of SMART and indicates a reasonable stability of the calibrations.

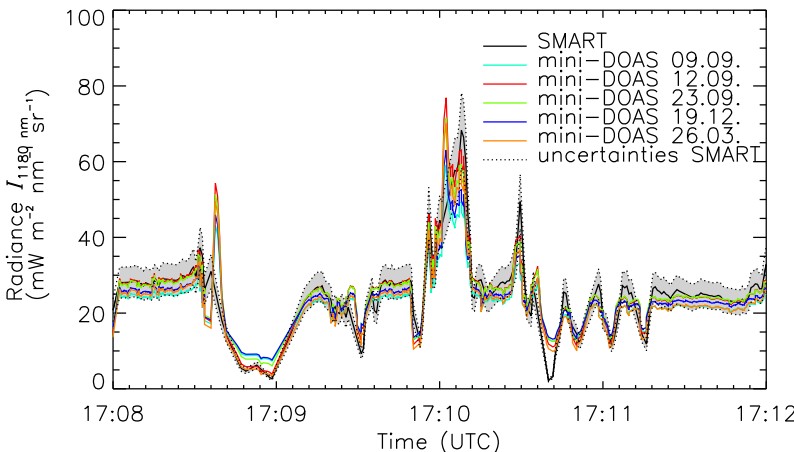

**Figure 10.** Time series of the nadir radiance of SMART $I^N_{S,1180}$ and of the mini-DOAS $I^N_{mD,1180}$ nadir channel at $\lambda = 1180$ nm using different calibrations as indicated in the legend. The uncertainty range of SMART radiance is shaded gray.

## 5  Retrieval of cirrus optical thickness

### 5.1  Iterative algorithm

By using all three calibrated radiance data sets obtained from SMART $I^N_S$, mini-DOAS nadir channels $I^N_{mD}$, and sideward viewing channels $I^V_{mD}$, an iterative retrieval algorithm of $\tau$ is developed and applied. It is based on the bi-spectral reflectance method described by Twomey and Seton (1980), and Nakajima and King (1990). Here, the retrieval is adapted for ice clouds with respect to ice crystal shape and used wavelengths, e.g. by Ou et al. (1995) and Rolland et al. (2000). For retrieving $\tau$ rough aggregates are assumed using pre-calculated ice crystal parametrizations after Yang et al. (2013). The iterative algorithm utilizes the spectral reflectivity $\mathcal{R}_\lambda$ which is defined as the ratio of spectral upward $I_\lambda$ to spectral downward $F^\downarrow_\lambda$,

$$\mathcal{R}_\lambda = \frac{I_\lambda \pi}{F^\downarrow_\lambda} \tag{3}$$

For the ML-CIRRUS data, $F^\downarrow_\lambda$ is taken from the actual SMART measurements on HALO. Measured $F^\downarrow_\lambda$ allows to identify and eliminate any influence of the radiation field above the aircraft, for example by cirrus. As an alternative to pre-calculate Look-up-Tables (LUT) by extensive forward simulations, an iterative algorithm is applied that runs RTS adjusted to each single measurement. This allows to set up simulations by actual input parameters for each measurement e.g. $\theta_0$, $\phi$, longitude, latitude and flight altitude. In that way, uncertainties caused by inaccurate assumptions in the RTS input are minimized. Additionally, the iterative method is not limited to a specific pre-calculated grid of $\tau$ and $r_{eff}$ as used in LUTs where a certain interval of preselected $\tau$ and $r_{eff}$ are given. The iterative algorithm automatically adjusts the range of $\tau$ and $r_{eff}$ without interpolation until

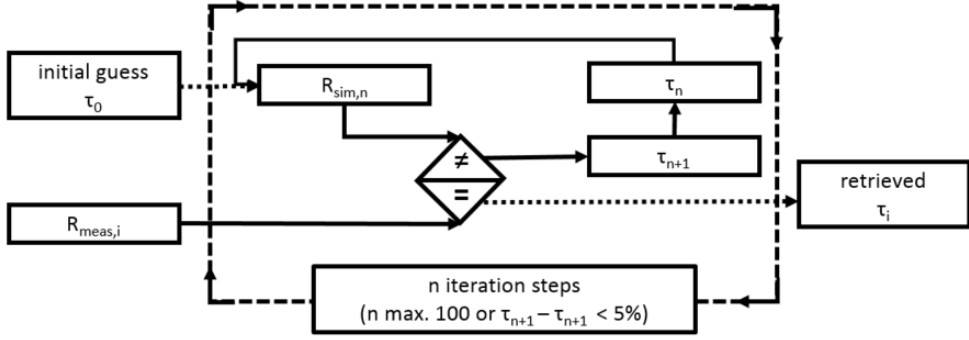

**Figure 11.** Scheme of the iterative algorithm. For every single measurement $i$ an iteration loop is started with an initial guess $\tau_0$ until measured $\mathcal{R}_{\mathrm{meas}}$ and simulated $\mathcal{R}_{\mathrm{sim}}$ reflectivity converge within $5\%$ difference or a maximum of 100 iteration steps is reached. At the end of the process the result is saved.

reaching the final result.

Figure 11 shows a scheme of the retrieval algorithm, which starts with an initial guess of $\tau_0$. Using the initial guess of $\tau$ and of any other cloud parameters, the cloud reflectivity $\mathcal{R}_{\mathrm{sim}}$ is simulated and compared to the measurements $\mathcal{R}_{\mathrm{meas}}$ of SMART and mini-DOAS, respectively. The ratio between $\mathcal{R}_{\mathrm{sim,n}}$ and $\mathcal{R}_{\mathrm{meas}}$ derived for each iteration step $n$ is used to scale the particular guess $\tau_{\mathrm{n}}$ by

$$\tau_{\mathrm{n+1}} = \tau_{\mathrm{n}} \cdot \frac{\mathcal{R}_{\mathrm{sim}}}{\mathcal{R}_{\mathrm{meas}}}. \tag{4}$$

The adjusted $\tau_{\mathrm{n+1}}$ is used in the RTS for the new iteration step $n+1$. The iteration of $\tau$ is repeated until the change of $\tau_{\mathrm{n}}$ between two iteration steps is smaller than $5\%$ or a limit of $n > 100$ iteration steps is reached. These stop criteria determine the accuracy of the iterative retrieval. If a lower relative stop criteria (change of $\tau_{\mathrm{n}}$ smaller than $5\%$ between two iteration steps or more then 100 iteration steps) is used the iteration may come closer to the true searched value and the retrieval accuracy increases as well as the necessary iteration steps and the computational time. To limit the computational time, the second stop criteria is used to limit the maximum number of iteration steps. For a typical cirrus observed during ML-CIRRUS with an average $\tau$ of 0.32, the cirrus optical thickness can be retrieved with a accuracy of about $\tau \pm 0.03$. The retrieval of $\tau$ by SMART and mini-DOAS bases on the measurements at $\lambda = 1180\,\mathrm{nm}$ and is scaled to $\lambda = 532\,\mathrm{nm}$ to consider the wavelength dependence of $\tau$ and to be able to compare it with WALES measurement at $\lambda = 532\,\mathrm{nm}$. Therefore, the retrieval considers RTS at both wavelengths. In the RTS $\tau$ is defined and changed at $\lambda = 532\,\mathrm{nm}$ while the measurements are compared to simulations at $\lambda = 1180\,\mathrm{nm}$ to determine the correct solution.

In case of measurements of optically thin cirrus, the retrieval can be applied for $\tau$ only. For these situations $I^{\mathrm{N}}_{\mathrm{RTS,1600}}$ at $\lambda = 1600\,\mathrm{nm}$ wavelength (ice absorption band) is too low and only measured with high uncertainty to retrieve $r_{\mathrm{eff}}$. For a cirrus cloud with $\tau = 0.03$, the simulated upward nadir radiance $I^{\mathrm{N}}_{\mathrm{RTS,1600}}$ and the sideward viewing radiance $I^{\mathrm{V}}_{\mathrm{RTS,1600}}$ in the range of $0.2\,\mathrm{mWm}^{-2}\mathrm{sr}^{-1}$. Such low $I$ are in the range of the electronic noise of the spectrometers leading to low Signal-to-Noise-Ratio

and high retrieval uncertainties. Especially for cirrus with low $\tau$ the variation of $I_{\mathrm{RTS,1600}}^{\mathrm{N}}$ and $I_{\mathrm{RTS,1600}}^{\mathrm{V}}$ with respect to changes in $r_{\mathrm{eff}}$ is low.

Simulations show, that for $\tau = 0.5$ the difference of $I_{\mathrm{RTS,1600}}^{\mathrm{N}}$ in nadir direction is only $0.1\,\mathrm{mW}$ when changing $r_{\mathrm{eff}}$ from $10\,\mu\mathrm{m}$ to $20\,\mu\mathrm{m}$ indicating the low sensitivity of $r_{\mathrm{eff}}$ retrievals at this wavelength. Therefore, a reliable retrieval of $r_{\mathrm{eff}}$ with reason-
able accuracy is not feasible. For $I_{\mathrm{RTS,1600}}^{\mathrm{V}}$ the difference is $1.4\,\mathrm{mWm^{-2}sr^{-1}}$ and about a magnitude larger indicating that a retrieval of $r_{\mathrm{eff}}$ might be reasonable. However, in order to be consistent between both nadir and sideward viewing retrieval, $r_{\mathrm{eff}}$ has been fixed. A value of $r_{\mathrm{eff}} = 30\,\mu\mathrm{m}$ was chosen, a typical value of ice crystals observed by in-situ measurements during ML-CIRRUS (Voigt et al., 2016). Therefore, the influence of an invalid assumption of $r_{\mathrm{eff}}$ on the iterative retrieval is analyzed. For this purpose the retrieval is tested for a typical cirrus of $\tau = 0.3$ and is run with three different assumptions of $r_{\mathrm{eff}}$ of $20\,\mu\mathrm{m}$,
$30\,\mu\mathrm{m}$, $40\,\mu\mathrm{m}$, representing the uncertainty of $r_{\mathrm{eff}}$. These simulations imply that the retrieved $\tau$ changes only by $\pm 0.02$ between smallest and largest $r_{\mathrm{eff}}$, resulting in a relative error in $\tau$ of $6.7\,\%$. The uncertainty in measured $I_{\mathrm{S,1600}}^{\mathrm{N}}$ and $I_{\mathrm{mD,1600}}^{\mathrm{V}}$ causes a retrieval uncertainty of less than $\tau \pm 0.2$. This justifies the fixed choice of $r_{\mathrm{eff}}$ in this specific cloud case.

However, the dependence of retrieved $\tau$ and the assumption of $r_{\mathrm{eff}}$ may vary with $\alpha$, ice crystal size, $\tau$ and $\lambda$ used in the retrieval.

## 5.2 ML-CIRRUS case study

The iterative retrieval is applied for a selected leg of the ML-CIRRUS flight on 26 March 2014. For this day the Terra MODIS image (overpass time 10:40 UTC) indicates clouds, with a west to east gradient in $\tau$ ranging from 5.8 to 0.38 (Fig. 12) including small cloud free regions. For large areas, cirrus with $\tau \leq 1$ is indicated by MODIS providing provides a well suited test case to compare sideward viewing and nadir observations even when $\tau$ ranges above the SVC level. As discussed in Section 2, for low
$\tau$ ranging up to 1, $\varepsilon_\tau$ of sideward viewing observations is higher than for nadir observations. An advantage of using a test case with $\tau$ higher than SVC is the insensitivity of the retrieval uncertainty with respect to the radiance measurement uncertainty. The reflected $I$ is still sufficiently large and exceeds the noise level of the nadir looking instruments to make a comparison between nadir and sideward viewing instruments possible.

In Figure 12 the flight track of HALO is indicated by the blue line. The cloud retrieval is applied to the HALO flight segment
for the leg between 08:15 UTC and 08:36 UTC (highlighted in red) when HALO did fly above the cirrus. During this period the aircraft flew constantly at 12.6 km height from South to North along $14°$ W. Due to low horizontal advection and hence slow cloud formation it can be expected that the Terra MODIS image (Fig. 12) actually reflects the cloud cover investigated by HALO. The cirrus developed along a warm conveyor belt and contained embedded contrails as indicated by the lidar backscatter profiles at $\lambda = 1064$ nm and $\lambda = 532$ nm of WALES (see Fig. 13). The time period for which $\tau$ is retrieved is marked
by the black frame. The selected flight segment is characterized by a constant cloud top height and a slightly increasing cloud bottom height towards northern flight direction. While the upper most cloud top is relatively homogeneous, there is significant variability in the layer below which is visible in the backscatter profile of WALES. The beginning of the black marked area shows high backscatter ratios of up to 500 indicating high reflectivity of a dense cirrus. At the end of the selected time period

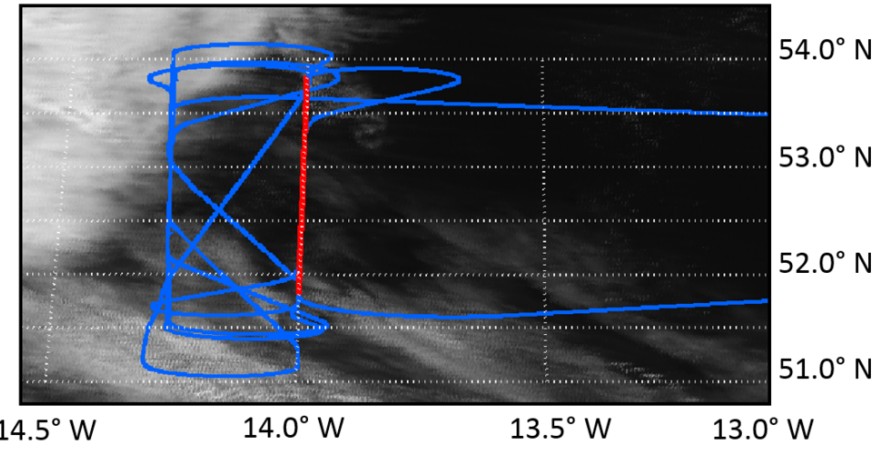

**Figure 12.** Investigated cloud field observed by MODIS-Terra on 26 March 2014. The flight track of HALO is indicated by the blue line. The flight leg between 08:15 UTC and 08:36 UTC for which the cirrus retrieval is performed is indicated by the red line.

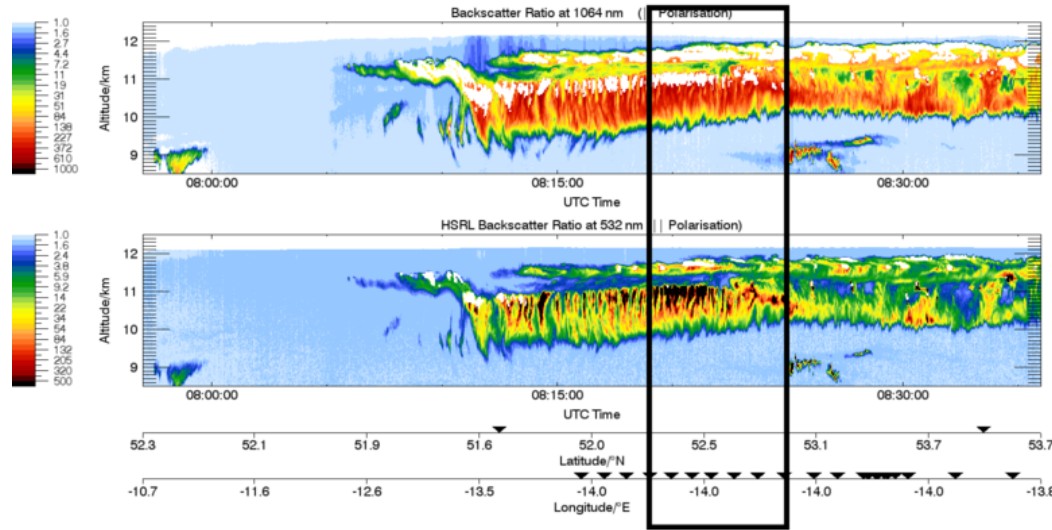

**Figure 13.** Vertical profiles of backscatter ratios at $\lambda = 1064$ nm (upper panel) and $\lambda = 532$ nm (lower panel) measured by WALES between 07:50 UTC and 08:50 UTC. The time period for which $\tau$ is retrieved is marked by the black rectangle.

the backscatter decreases. The lower part of the cirrus shows small-scale variability mainly connected to sedimentation of ice crystals.

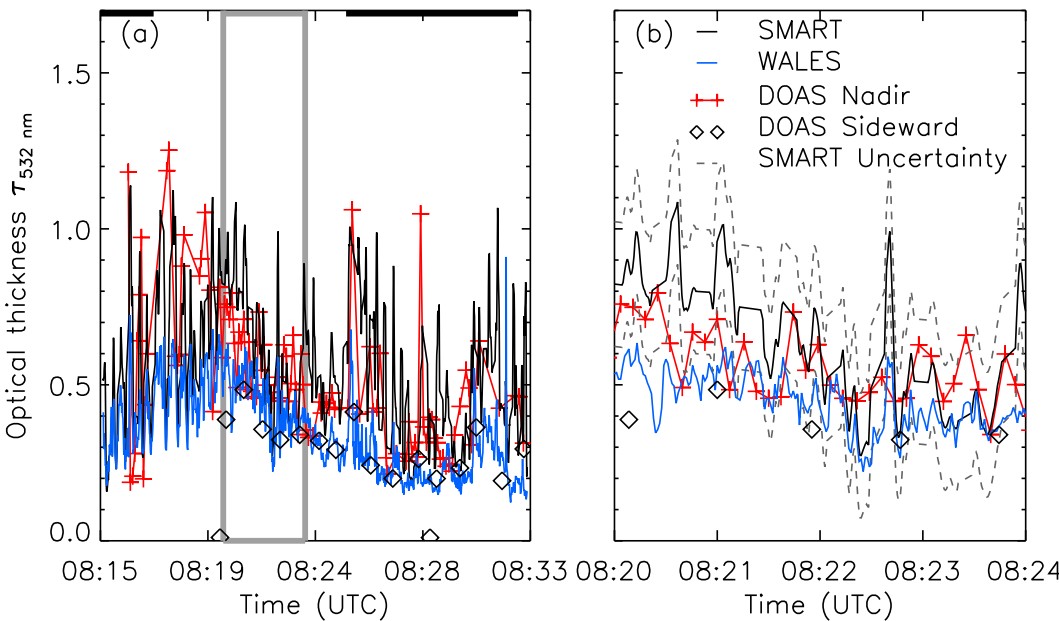

**Figure 14.** Time slices of the investigated flight segment on 26 March 2014 (a) and zoom (b) of $\tau$ at $\lambda = 532$ nm retrieved from SMART (black line), WALES (gray line), mini-DOAS sideward viewing (diamonds) and nadir spectrometers (crosses) along the flight track of ML-CIRRUS flight on 26 March 2014. Periods with the second cloud layer are marked by the black lines at the top of (a).

### 5.2.1 Time series of cirrus optical thickness

Figure 14a shows a 20 minutes long flight segment of retrieved $\tau$ at $\lambda = 532$ nm calculated from SMART, mini-DOAS nadir and sideward viewing spectrometers. WALES measurements are included for comparison. Along the analyzed cirrus, the retrieved $\tau$ ranges between 0.1 and 1.3 indicating the horizontal variability of the cirrus. The general decrease of $\tau$ towards
5  higher latitudes (increasing time) matches with the cloud pattern observed by WALES. While SMART and mini-DOAS nadir channels resolve the cirrus variability observed by WALES, the sideward viewing channel retrieval does not cover these fluctuations due to the reduced time resolution of the scanning mode and the large spatial scale (tenth of kilometers) over which sideward viewing measurements average. At some locations, e.g. 08:21 UTC, $\tau$ retrieved by SMART and mini-DOAS significantly exceed the measurements of WALES. Mostly likely both instruments retrieve larger $\tau$ than WALES since ice crystals
10  were falling out of the cirrus obscured to the Lidar measurements. A second segment with higher retrieved $\tau$ is likely due to an underlying cirrus between 8.5 km and 9.5 km altitude that is also obscured to the detection by WALES. Therefore, a positive systematic offset of the retrieved $\tau$ occurs for SMART and mini-DOAS. These data points are excluded from the following analysis. Nevertheless, there is a slight chance that a few cloud fragments of these second cloud layers are still affecting the SMART- and mini-DOAS retrieval. Both passive sensors have a larger FOV compared to WALES and, therefore, are more

likely sensitive to cloud layers located below the cirrus.

Average $\tau$ are calculated for the filtered time period (indicated by the grey box in Fig. 14) for each instrument. Due to different sampling intervals, a different resolution and number of observations are included in the averaging calculations. The retrieved average of $\tau$ at 532 nm is $0.54 \pm 0.2$ (SMART), $0.49 \pm 0.2$ (mini-DOAS nadir spectrometer), $0.27 \pm 0.2$ (mini-DOAS side-
ward viewing spectrometer) and $0.32 \pm 0.02$ (WALES). The results indicate a reasonable agreement of $\tau$ retrieved by SMART and mini-DOAS nadir channel, while lower $\tau$ are inferred from mini-DOAS sideward viewing and WALES measurements. Taking the WALES measurements as a reference, the measurements of SMART and mini-DOAS overestimate $\tau$. However, by estimating the uncertainty of the mini-DOAS and SMART basing on RTS, the measurement error of $I^{\mathrm{N}}_{\mathrm{S},1180}$ (14.5 %) by SMART results in an uncertainty range of retrieved $\tau$ of $\pm 0.2$, which covers the values of $\tau$ obtained by WALES. The un-
certainty range of $\tau$ is determined by running the retrieval twice with a bias of measured $I^{\mathrm{N}}_{\mathrm{S},1180}$ with $\pm 14.5$ % uncertainty at 1180 nm wavelength as upper and lower border. The resulting upper and lower retrieved $\tau$ represent the retrieval uncertainty. The mean $\tau$ inferred from the mini-DOAS sideward viewing observations is significantly lower than measured by SMART and mini-DOAS nadir measurements. Differences in $\tau$ range up to $\pm 0.73$ between SMART and mini-DOAS sideward viewing observations. This may result from the different FOV of the sideward viewing geometry that does not observe the exact same
clouds as SMART and nadir channels did. With the scanning sensors orientated to starboard the sideward viewing retrieval corresponds to cirrus 8 km east of the flight track. As the MODIS satellite image in Fig. 12 indicates, the cirrus becomes slightly thinner towards east, which possibly is due to the lower values of $\tau$. Other potential reasons are the assumed ice crystal shapes for the RTS and different field-of-view of the passive and active remote sensing instruments. On the other hand, the agreement between mini-DOAS sideward observations and WALES is significantly better. The maximum difference of $\tau$ be-
tween mini-DOAS sideward channels and WALES is $\pm 0.25$ while the difference between the mean values is $\pm 0.05$ (15.6 %). With WALES and mini-DOAS measuring in different viewing geometries but showing better agreement, the differences of $\tau$ retrieved by SMART is most likely caused by uncertainties in $\alpha$. As discussed in Section 2.3, nadir observations are stronger affected by $\alpha$ than sideward observations. This is confirmed by the smaller differences between WALES and mini-DOAS sideward observations and indicates the advantage of the sideward viewing retrieval due to a reduced surface influence and lower
retrieval uncertainty.

Figure 14b displays a zoom of the time series between 08:20 UTC to 08:24 UTC. During this flight segment, $\tau$ inferred by WALES is characterized by systematic oscillations varying between 0.2 and 1.1 also visible in the backscatter profile of WALES in Fig. 13. The lag time between two maxima is approximately between 20 s and 25 s flight time, which corresponds to a horizontal distance between 4.4 km and 5.5 km. This pattern is present in the measurements of SMART, WALES and the
mini-DOAS nadir channels even though partly obscured in the latter measurements due to its reduced time and space resolution.

Figure 15a to d show scatter plots of retrieved $\tau$ for the different instrument combinations. A linear regression through the origin is performed and displayed in all cases. Data where a second cloud layer was present below the cirrus (gray points) are excluded. The comparison between SMART and WALES in Fig. 15a shows that the majority of the data is below the 1:1 line
(gray). The linear regression results in $f(x) = 0.6621 \cdot x$. The regression confirms that SMART systematically retrieves higher

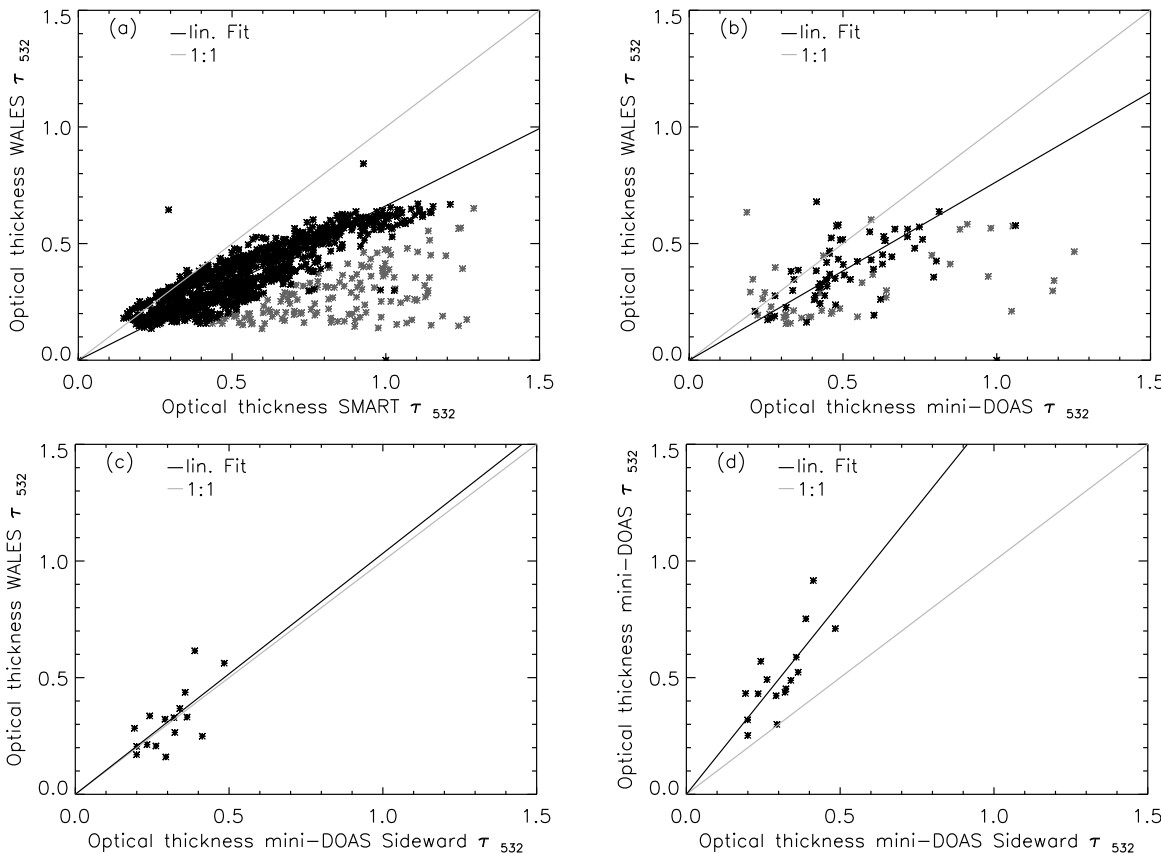

**Figure 15.** (a) Comparison of the retrieved cirrus optical thickness $\tau$ by WALES and SMART at $\lambda = 532$ nm wavelength. (b) Comparison of the retrieved cirrus optical thickness $\tau$ by WALES and mini-DOAS nadir channel at $\lambda = 532$ nm wavelength. Measurements when a second cirrus layer was present are displayed in grey and are discarded in the regression. (c) Comparison of the retrieved cirrus optical thickness $\tau$ by WALES and mini-DOAS sideward viewing channels at $\lambda = 532$ nm wavelength. No data is discarded. (d) Comparison of the retrieved cirrus optical thickness $\tau$ by mini-DOAS nadir and sideward viewing channels at $\lambda = 532$ nm wavelength.

$\tau$ compared to WALES.

Compared to SMART, mini-DOAS nadir observations of $\tau$ depart less from WALES (Fig. 15b. Similar to SMART, the slope of the linear fit $f(x) = 0.6943 \cdot x$ indicates that mini-DOAS systematically overestimates $\tau$ compared to WALES. This similarity between SMART and mini-DOAS is obvious as SMART and mini-DOAS rely on the same radiometric calibration and
5  retrieval. As indicated in Fig. 14b retrieved $\tau$ from WALES and the mini-DOAS sideward viewing channels agree well confirmed by the linear regression in Fig. 15c that gives a slope of $f(x) = 1.0328 \cdot x$ close to unity. The overestimation of retrieved $\tau$ by the mini-DOAS nadir channels compared to the sideward channels is visible in Fig. 14d which results in a linear fit of $f(x) = 1.642 \cdot x$. Overall the comparison provides evidence that the inferred $\tau$ agrees between the different sensors.

Having nadir and sideward viewing observations at the same time allows to select the appropriate measurement geometry

depending on cloud situation, e.g. $\tau$ and $\alpha$. The sensitivity studies in Section 2.4 suggest that a combination of nadir and sideward viewing measurements allow a retrieval of $\tau$ for wide range of cirrus clouds depending on the observation conditions. For thin clouds the sideward viewing geometry would be preferred. In case the cloud becomes optically too thick, leading to high upward $I^{\mathrm{V}}_{\mathrm{S},1180}$ and a saturation of $\varepsilon_\tau$, no retrievals of $\tau$ are possible. Then, switching to nadir observations of $I^{\mathrm{N}}_{\mathrm{S},1180}$ still

enables to determine the amount of reflected radiation and to retrieve $\tau$.

### 5.2.2   Probability distribution of cirrus optical thickness

For further comparison the probability density functions (PDF) of $\tau$ retrieved by SMART, mini-DOAS nadir spectrometers and WALES was investigated. A PDF of mini-DOAS sideward viewing spectrometers is not included because of the limited

number of data points making a statistically meaningful PDF impossible. The PDF are shown in Fig. 16. Corresponding mean and median values of the distributions are given in Tab. 4. SMART (black solid line) and mini-DOAS (red solid line) which base on the same radiometric calibration and retrieval method show a comparable PDF indicating that both instruments measured the same cloud area. In both cases observed $\tau$ range from 0.15 to 1.25 for SMART and mini-DOAS and from 0.15 to 0.7 for WALES (black dashed line). The PDF maxima for SMART and mini-DOAS is around $\tau = 0.4$, slightly more pronounced

for the mini-DOAS. For SMART and mini-DOAS, the PDF are skewed to small $\tau$ with a median of 0.47 for SMART and 0.48 for mini-DOAS. This is slightly smaller than the mean value of 0.5 for SMART and 0.51 for mini-DOAS. Both PDF are long-tailed towards large $\tau$, slowly decreasing to higher values of $\tau$. In contrast, $\tau$ measured by WALES (black dashed line) shows a stronger shift to low $\tau$ as the mean value of $\tau$ is significantly lower. The most frequent $\tau$ is around 0.2. The WALES measurement do not show $\tau$ larger than 0.7. This results in a stronger decrease of the WALES PDF to higher $\tau$ compared

to SMART and mini-DOAS. The difference may be explained by different FOV and therefore measuring different horizontal parts of the clouds. It is assumed, that SMART and mini-DOAS, e.g. due to similar large FOV, average over larger areas and are influenced by 3-D radiative effects caused by clouds, atmosphere and surface, which are not considered in the presented 1-D RTS and the iterative retrieval (Davis et al., 1997). Contrarily WALES has a more narrow FOV resulting from an opening angle of the telescope of $0.08°$. Because of the smaller FOV of WALES the spot of the laser at cloud top covers a smaller

area compared to SMART and mini-DOAS which have a spatial resolution in the range of tenth of meters depending on the distance between aircraft and cloud top. Therefore, WALES resolves finer cloud structures that may exhibit lower $\tau$ (cloud gaps) or larger $\tau$. In case of the most unfortunate situation WALES would measure a cloud free region but SMART and mini-DOAS receive $I^{\mathrm{N}}$ from a much larger area including clouds with various $\tau$. This better spatial resolution of WALES to SMART and mini-DOAS may explain the shift of WALES to lower $\tau$ but does not give reasons for the lower amount of high $\tau$.

Differences in the PDF of $\tau$ may also result from the measurement methodologies. While WALES uses a laser with small FOV for active remote sensing, SMART and mini-DOAS are passive remote instruments relaying on scattered sunlight. Therefore, SMART and mini-DOAS are influenced by the RTS of the whole atmosphere, while WALES is only sensitive to scattering within its narrow LOS. Additionally, the different wavelengths of the measurements may introduce biases in the retrieved $\tau$ due to different penetration depth of the reflected radiation into the cloud (Platnick, 2000). Therefore, the wavelength selection

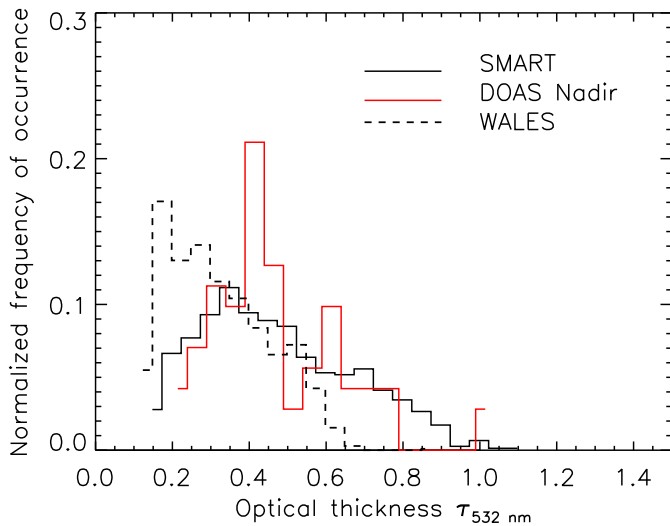

**Figure 16.** PDFs of cirrus optical thickness $\tau$ at $\lambda = 532$ nm retrieved from SMART, mini-DOAS and WALES measurements. The bin size is 0.05 units of $\tau$.

defines the layer in the cloud which is probed. While WALES uses backscatter measurements at $\lambda = 532$ nm and $\lambda = 1064$ nm the measurements of $I_{S,1180}$ by SMART and mini-DOAS are performed at $\lambda = 1180$ nm. Although the retrieval accounts for the wavelength dependence of scattering, absorption and refraction on ice crystals (Takano and Liou, 1989; Yang et al., 2013) by scaling the retrieved $\tau$ at $\lambda = 1180$ nm to $\lambda = 532$ nm to make it comparable between the different instruments.

Referring to the sensitivity studies from Section 2 the influence of $\alpha$ and the ice crystal shape effects on the upward $I$ measured in nadir geometry is larger compared to the sideward viewing measurements. While nadir observations, especially of optical thin clouds, are strongly influenced by $\alpha$, sideward viewing observations are less effected. This is demonstrated in this case study where the sea surface albedo may vary due to different surface wind speeds (Cox and Munk, 1954) and indicates the advantage of sideward viewing measurements. An other possible reason for the differences in the PDF and the mean values

between mini-DOAS nadir and sideward retrievals of $\tau$ are the varying angular dependencies of measured $I$ for different ice crystal shapes. For the RTS in the retrieval an assumption for the ice crystal shape has to be made which slightly influences the result for the nadir retrieval. This is more pronounced for the retrieval using the sideward channels of the mini-DOAS which is presented in the sensitivity study in Section 2.2. The WALES measurements are less effect by different ice crystal shapes but more on the ice crystal size assumption which is a general difference between the active and remote sensing instruments

presented here.

**Table 4.** Mean and median of the PDFs of cirrus optical thickness $\tau$ derived from WALES, SMART and mini-DOAS.

|  | mean | median |
|---|---|---|
| WALES | 0.35 | 0.33 |
| SMART | 0.56 | 0.52 |
| mini-DOAS | 0.52 | 0.47 |

## 6 Conclusions

The potential of airborne spectral radiance measurements in sideward viewing direction for cirrus remote sensing is investigated. For this purpose radiative transfer simulations (RTS) are performed and airborne measurements of the Spectral Modular Airborne Radiation measurement sysTem (SMART) and the Differential Optical Absorption Spectrometer (mini-DOAS) are
compared. A sensitivity study based on RTS showed that sideward viewing measurements are generally more suited for detecting and investigating optically thin cirrus than observations in nadir orientation. Using sideward viewing observations the sensitivity $\varepsilon_\tau$ of measured radiance $I^V$ is larger than for nadir measurements up to a factor of ten depending on the selected observation geometry and cloud properties. For cirrus optical thickness $\tau \leq 1$ and all simulated sideward vieweing geometries $\varepsilon_\tau$ is larger compared to nadir observations. This results in a higher retrieval accuracy due to a reduced influence of measure-
ment uncertainties. The RTS indicate that large observation angles $\theta_V$ (close to the horizon) and small relative solar azimuth angle $\phi$ (observations in direction of the Sun) result in highest $\varepsilon_\tau$.

For retrievals of $\tau$ using sideward viewing measurements, the wavelength selection is crucial. Simulations indicate that wavelengths larger than $\lambda = 900$ nm are best suited. Reflected $I^V$ of smaller wavelengths is significantly contaminated by scattering and absorption due to the reducing interference from Rayleigh scattering. Furthermore, the sideward viewing orientation re-
duces the influence of the surface albedo $\alpha$ on reflected $I^V$. As a result, a precise assumption of $\alpha$ in the retrieval algorithm is less crucial. This substantially improves the uncertainties of passive solar remote sensing especially in locations of highly variable $\alpha$, where an exact assumption of $\alpha$ is impossible.

Contrarily, for sideward observations, a reasonable good assumption of the ice crystal shape used in the RTS is important. The RTS showed that in sideward viewing geometry the shape effects on reflected $I^V$ are more pronounced than for nadir measure-
ments. An incorrect assumption would bias the retrieval of $\tau$ significantly. On the other hand, the sensitivity for different ice crystal shapes may offer the possibility to retrieve shape information when measuring at different viewing angles. Nevertheless, smoothing of horizontal variability of optical thickness-fields by sideward viewing observations has to be taken into account.

Using the SMART, mini-DOAS nadir and sideward measurements in conjunction with an iterative retrieval, $\tau$ is derived for a case study of ML-CIRRUS. The inferred $\tau$ from SMART, mini-DOAS and the additional lidar measurement by the Water
Vapour Lidar Experiment in Space (WALES) show a reasonable good agreement in $\tau$ for the nadir channels with absolute differences of $\pm 0.22$ (66.6 %) between SMART and WALES and $\pm 0.17$ (52.3 %) between mini-DOAS and WALES observations respectively. The retrieval using mini-DOAS sideward channels is also successful demonstrated for a reduced set of

observations limited to $\theta_V$ between $85°$ and $90°$. Differences in $\tau$ range up to $\pm 0.73$ between SMART and mini-DOAS sideward viewing observations and are partly caused by the different viewing geometries. First, the sideward telescopes view into starboard direction, probing the cirrus cloud top at approximately 8000 m aside the flight track. Second, the nadir observations may suffer from uncertainties in $\alpha$ while the sideward observations are less effected by changes in $\alpha$. Even for sea surfaces

as presented here, $\alpha$ may change due to different wind speeds. Other potential reasons are the assumed ice crystal shapes in the RTS and different field-of-view of the passive and active remote sensing instruments. This conclusion is apparent from different probability distributions. While SMART and mini-DOAS show a median around $\tau = 0.4$, the median for WALES is shifted to lower $\tau$ around 0.2, indicating that WALES observed small $\tau$ more frequently. The difference of mean values of $\tau$ between mini-DOAS sideward channels and WALES is smaller with $\pm 0.05$ ($15.6\%$). This shows the advantage of the sideward

viewing retrieval due to a reduced surface influence and lower retrieval uncertainty, because of high $\varepsilon_\tau$ compared to the nadir measurements. For future dedicated cloud observations it is recommended to adjust $\theta_V$ to the most sensitive direction between $60°$ and $90°$ to reduce the uncertainty in the sideward viewing retrieval. Additional sideward viewing scans in homogeneous cloud conditions might be used to estimate the cirrus ice crystal shape and minimize the retrieval uncertainties. The case study shows that cirrus retrieval using airborne sideward viewing observations with mini-DOAS are possible and can increase the

potential of remote sensing on HALO significantly. Therefore, we suggest sideward viewing measurements for passive remote sensing of optically thin cirrus clouds.

*Acknowledgements.* This research was funded by the German Research Foundation (DFG, HALO-SPP 1294). The authors acknowledge the support by the Deutsche Forschungsgemeinschaft (DFG) through grants PF 384/7-1/2, PF 384/16-1 and WE 1900/35-1. The authors thank the pilots and appreciate the support by the Flugbereitschaft of the German Aerospace Center (DLR). Additionally the authors thank

enviscope GmbH for preparation and testing of SMART.

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
