# Peer review of "Potential of remote sensing of cirrus optical thickness by airborne spectral radiance measurements in different sideward viewing angles"

_Atmospheric Chemistry and Physics, 2016_

## Referee Comment (RC1) · Anonymous Referee #1 · 17 Nov 2016

The paper "Potential of remote sensing of cirrus optical thickness by airborne spectral radiance measurements in different viewing angles and nadir geometry" presents a very interesting sensitivity analysis of the influence of different parameters in the measured upward radiance using radiative transfer simulations in presence of cirrus. Results from in-situ aircraft measurements using SMART and a scanning mini-DOAS are also presented. The paper is suitable for publication in ACP after some minor corrections and improvements are performed. See detailed comments below:

In general, I don't see a clear connection between the sensitivity analysis presented in

Section 2 and the results and discussion presented in Section 4. Some paragraphs or sentences explicitly linking the two sections where necessary would be appropriate.

Abstract, lines 17-18. The simulations indicate that off-nadir measurements are more adequate to retrieve  $\tau$  of thin clouds, but that is not observed in the retrievals from the aircraft measurements presented here (at least in the way they are currently presented). Please, rephrase.

Page 2, line 19. "better quantify" instead of "quantify better". It is not clear what you mean by "appear worthwhile", rephrase.

Page 3, line 1. Add a comma after "relevant parameters"

Page 3, line 5. Elaborate more the statement "As a result, airborne remote sensing is required to bridge local in-situ and global satellite observations."

Page 3, line 20: "and are not routinely be used in trace gas measurements" is not clear. Please, rephrase.

Page 5, line 5. The use of the acronym SZA and the symbol  $\theta$ 0 for the solar zenith angle is redundant. Remove the acronym.

Page 6, figure 2. In the lower part of the figure it will be more convenient to plot the relative differences normalized to the Radiance. That will help with the corresponding discussion in lines 13-16. Also, some text is missing in the figure caption.

Page 6, line 3. Replace "wavelengths less..." by "wavelengths lower..."

Page 7, lines 4-5. "The RTS suggest that off-nadir observations at near infrared wavelengths ( $\tau$ > 900 nm) are more suitable for the detection of SVC and cirrus."

Page 8, figure 4 and lines 9-13. Because of the different values of I under the different constraints you should consider providing the sensitivity in percentages.

Page 9, line 2. Do you mean "thick clouds, for larger optical thickness..." here?
Page 9, line 13. Remove "especially"

Page 9, line 25. You should consider include a plot with the steepest derivative  $\gamma$  (maybe a subplot in Figure 5?)

Page 11, figure 6. Please, include a subplot with the relative differences between the different ice crystals. This will help with the discussion in lines 8-13.

Page 12, line 6. "were investigated"

Page 12, line 11. Provide references for SMART and the calibration procedure.

Page 12, line13-14. Provide references for the mini-DOAS and the DOAS technique.

Page 13, line 21. The symbol ILmD has not been defined before. Please, define.

Page 14, line 26. Why are multiple scattering effects neglected?

Page 16, Figure 8. Can you add the error bars to the plots? Especially to plots b and d. Idem for figure 9.

Page 19, lines 26-27. Please, elaborate the statement "These stop criteria determine the accuracy of the iterative retrieval."

Page 20, lines 1-15. What happens for off-nadir observations?

Page 21, Figure 12. Axis labels are missing.

Page 22, lines 14-15. Are these average values obtained for the coincident measurements only? Otherwise, comparing the different values is not realistic. Especially for the DOAS off-nadir, which have a smaller temporal resolution and does not capture all the variability observed during the analyzed period.

Page 23, line 3. A more in-depth analysis of the uncertainty will be useful, mainly for inter-comparison purposes between the different datasets presented in figure 14.

Page 23, line 9 and figure 14. It looks like there is a better agreement between the
DOAS off-nadir and the reference WALES than between the DOAS off-nadir and DOAS nadir or SMART. Can you comment something on that? Can you further discuss the advantages and disadvantages of having nadir and off-nadir measurements and link it with the sensitivity analysis in section 2?

Page 23, lines 20-21. This statement is not clear. If the data points contaminated by the second cloud layer are excluded from the calculations, what do you mean here?

Page 24, lines 10-12. This is not clear either. From the results and the discussion presented before, it looked like you were using the wavelength of 532 nm for all the instruments. Please, clarify where necessary.

Page 26, line 14. Agreement is within the uncertainty but I would not consider a 66.6Numerical values for the differences between DOAS nadir and Wales and DOAS off-nadir should be included separately. Relevance was given to the comparison between the nadir and off- nadir observations in the sensitivity analysis and it will be interesting to do a clear distinction also for the in-situ airborne data and include a significant conclusion at this respect.

---

## Referee Comment (RC2) · Anonymous Referee #2 · 29 Dec 2016

This paper aims to understand the differences of nadir and off-nadir measurements for cirrus optical thickness retrievals. The simple simulations did show the differences between the nadir and off-nadir measurements. However, the observations failed to present the same picture rather than show significant differences between active and passive measurements. Thus, the paper is not coherently organized. Other than this major issue, the paper made many inconsistent statements or conclusions. Therefore, significant revisions are needed to make it publishable.

Page 2: there are significant discussions for SVC, but techniques presented here are

not suitable for dealing with cirrus with such small optical depth- large uncertainties among them.

Page 2, last sentence—it is not an accurate statement if you consider passive sensor measurements.

Page 3: Lines 5-6: the cirrus optical thickness of water clouds does not make sense—re-write.

Page 4: Line 18: If you conclude that it is impossible here. You don't need any further study in this paper. Yes, it is challenging, which indicates that we need more observational constrains to improve the retrieval.

Page 4, Lines 19-21: The statements here are not accurate. Off-nadir measurements are widely used for space-base cirrus remote sensing. As you know, most satellite passive sensors are wide swath measurements .

Page 4, Line 25, "highly sensitive": An overstatement. Yes, it is more sensitive, but it is highly dependent on the magnitude of off-angle.

Page 5, Line 15: What does "F" in "FDISORT" mean?

Page 6: Figure 2 caption in the PDF misses words.

Page 6, line 15: The statement of "cirrus can not be detected" is not accurate. Cirrus is a general category including high clouds with optical depth up to 3.

Page 7, lines 4-5: To draw this conclusion, you'd better to present calculation results with a higher optical depth.

Page 7, line 15: This statement does not consistent with the statements in the next paragraph.

Page 8, line 1-2: To draw this conclusion, you need to make many assumptions.

Page 8, line 7: Based on the statement, it seems that you don't consider angle smaller

than 60 degree as the off-nadir observations. That is not right.

Page 9, line 9-10: It is hard to understand this sentence.

Page 12, figure 7: It is hard to see the location of the optical port in (b). A better figure may be needed.

Page 12, line 12: UV and VIS were defined early.

Page 12, line 13: DOAS was defined early. –Avoid multiple definitions.

Page 15, line 3: "cross-calibrate both instrument" is no right. As you discussed in the paper, SMART is lab calibrated.

Page 16, lines 1-2: Giving absolute numbers are needed, but it will be good to present relative differences too.

Page 18, line 3: Based on Fig. 10, I'd like to say that 2.9 is a big number, which is difficult to support the stable calibration consistent.

Page 19, Line 26: The statement here is not consistent with the lowest box in Fig. 11.

Page 20, line 20: Even for lidar guy, it is hard to see contrails in Fig. 13. How about to plot Fig. 13 as a color figure to make the fine feature easy to identify.

Page 22, line 22: For cirrus cloud optical depth around 1, it is hard to claim that the lower layer is obscured by the upper cloud layer. The lower layer can be clearly identified from lidar image.

Page 23, line 3: Is 10% here mean error or random error? You need to explain the +0.2 overestimation.

Page 23, lines 16-23: Which kind of calibration errors explain the good linear correlations and 0.66 or 0.69 slopes?

Page 24, lines 10-11: For large ice crystals, why do you expect optical depth difference between 532 nm and 1180 nm?

---

## Author Comment (AC1) · 24 Feb 2017

We thank the reviewer for the encouraging words and fort he helpful comments which improved the manuscript noticeably. By adding some more explanations and hints from a person not involved in the manuscript preparation enhanced the understanding for the reader.

The replies of the reviewer comments are given in the following manner: Reviewer comments are printed in bold, are labeled, and are listed in the beginning of each answer. The reviewer comments are followed by the author comments and revised parts of the paper. The revised parts of the paper are written in quotation marks and italic letters.

**Comments:**

1. **In general, I don't see a clear connection between the sensitivity analysis presented in Section 2 and the results and discussion presented in Section 4. Some paragraphs or Sentences explicitly linking the two sections where necessary would be appropriate.**

   The reviewer is right, the connection between the sensitivity study and the application to airborne measurements was not exactly pointed out. This might have been caused by relative high optical thickness of the cirrus selected for the case study. The choice of this case is now justified in the manuscript. The cirrus ranges in the range of $\tau < 1$ where sideward viewing observations are more sensitive compared to nadir observations. The following passages have been added and linking sentences and references were added to the corresponding sections 2 and 4 to show the linkage between simulations and measurements. Additional, in several parts of Section 4 and 5 the revised manuscript now refers to the results of the sensitivity study and highlights the differences between the two viewing geometries.

   *"The RTS suggest that sideward viewing observations at near IR wavelengths ($\lambda > 900$ nm) are more suitable for the detection of SVC and cirrus. As a result the retrieval in Section 4 is performed at 1180 nm and 1600 nm wavelength in the IR region which are sensitive to tau and reff and not disturbed by Rayleigh scattering."*

   *"Considering these findings, the retrieval of tau in Section 4 is performed for $\Theta_V <= 60°$ only."*

   *"The sensitivity studies in Section 2.4 suggest that a combination of nadir and sideward viewing measurements allow a retrieval of tau for wide range of cirrus clouds depending on the observation conditions. For thin clouds the sideward viewing geometry would be preferred. In case the cloud becomes optically too thick, leading to high upward $I^V_{S,1180}$ and a saturation of $\varepsilon_\tau$, no retrievals of tau are possible. Then, switching to nadir observations of $I^N_{S,1180}$ still enables to determine the amount of reflected radiation and to retrieve $\tau$."*

   *"Referring to the sensitivity studies from Section 2 the influence of alpha and the ice crystal shape effects on the upward I measured in nadir geometry is larger compared to the sideward viewing measurements. While nadir observations, especially of optical thin clouds, are strongly influenced by $\alpha$, sideward viewing observations are less effected. This is demonstrated in this case study where the sea surface albedo may vary due to different*

*surface wind speeds (Cox and Munk, 1954) and indicates the advantage of sideward viewing measurements."*

*"The retrieval using mini-DOAS sideward channels is also successful demonstrated for a reduced set of observations limited to $\Theta_V$ between 85° and 90°. Differences in $\tau$ range up to +-0.73 between SMART and mini-DOAS sideward viewing observations and are partly caused by the different viewing geometries. First, the sideward telescopes view into starboard direction, probing the cirrus cloud top at approximately 8000 m aside the flight track. Second, the nadir observations may suffer from uncertainties in $\alpha$ while the sideward observations are less effected by changes in $\alpha$. Even for sea surfaces as presented here, alpha may change due to different wind speeds. Other potential reasons are the assumed ice crystal shapes in the RTS and different field-of-view of the passive and active remote sensing instruments. This conclusion is apparent from different probability distributions. While SMART and mini-DOAS show a median around $\tau=0.4$, the median for WALES is shifted to lower $\tau$ around 0.2, indicating that WALES observed small $\tau$ more frequently. The difference of mean values of $\tau$ between mini-DOAS sideward channels and WALES is smaller with +-0.05 (15.6%). This shows the advantage of the sideward viewing retrieval due to a reduced surface influence and lower retrieval uncertainty, because of high $\varepsilon_\tau$ compared to the nadir measurements."*

2. **Abstract, lines 17-18. The simulations indicate that off-nadir measurements are more adequate to retrieve _ of thin clouds, but that is not observed in the retrievals from the aircraft measurements presented here (at least in the way they are currently presented). Please, rephrase.**

This is right, in the original manuscript the focus of the discussion was more on the discrepancy between nadir and WALES measurements rather than highlighting the good agreement of sideward viewing observations and WALES. In the revised manuscript, the view of the reader is now more shifted to this good agreement, what indeed reflects the results from the sensitivity study:

*"The mean $\tau$ inferred from the mini-DOAS sideward viewing observations is significantly lower than measured by SMART and mini-DOAS nadir measurements. Differences in $\tau$ range up to +-0.73 between SMART and mini-DOAS sideward viewing observations. This may result from the different FOV of the sideward viewing geometry that does not observe the exact same clouds as SMART and nadir channels did. With the scanning sensors orientated to starboard the sideward viewing retrieval corresponds to cirrus 8 km east of the flight track. As the MODIS satellite image in Fig. 12 indicates, the cirrus becomes slightly thinner towards east, which possibly is due to the lower values of $\tau$. Other potential reasons are the assumed ice crystal shapes for the RTS and different field-of-view of the passive and active remote sensing instruments. On the other hand, the agreement between mini-DOAS sideward observations and WALES is significantly better. The maximum difference of $\tau$ between mini-DOAS sideward channels and WALES is +-0.25 while the difference between the mean values is +-0.05 (15.6%). With WALES and mini-DOAS measuring in different viewing geometries but showing better agreement, the differences of $\tau$ retrieved by SMART is most likely caused by uncertainties in $\alpha$. As discussed in Section 2.3, nadir observations are stronger affected by $\alpha$*

*than sideward observations. This is confirmed by the smaller differences between WALES and mini-DOAS sideward observations and indicates the advantage of the sideward viewing retrieval due to a reduced surface influence and lower retrieval uncertainty."*

*"As indicated in 14 (b) retrieved τ from WALES and the mini-DOAS sideward viewing channels agree well confirmed by the linear regression in Fig. 14 (c) that gives a slope of f(x) = 1.0328 * x close to unity. The overestimation of retrieved τ by the mini-DOAS nadir channels compared to the sideward channels is visible in Fig. 14 (d) which results in a linear fit of f(x) = 1.642 * x."*

3. **Page 2, line 19. "better quantify" instead of "quantify better". It is not clear what you mean by "appear worthwhile", rephrase.**

   *"In order to quantify the microphysical and optical properties of SVC, which are needed to determine their radiative effects, more observations of this cloud type are required."*

4. **Page 3, line 1. Add a comma after "relevant parameters"**

   Comma was added

5. **Page 3, line 5. Elaborate more the statement "As a result, airborne remote sensing is required to bridge local in-situ and global satellite observations."**

   We rephrased this section to point out the relevance of airborne measurements in comparison to satellite and ground based observations.
   *"While satellite observations are suited to study the global coverage of cirrus, their spatial and temporal resolution is still limited and can not resolve the high spatial variability of cirrus. As a consequence the 3-D radiative effects of different cirrus properties, e.g., tau, ice crystal size and shape, can not be studied using the coarse resolution of satellite remote sensing. Ground-based lidar and radar remote sensing can provide a high temporal resolution but are limited to a fixed location. In-situ airborne measurements can provide cirrus properties with both."*

6. **Page 3, line 20: "and are not routinely be used in trace gas measurements" is not clear. Please, rephrase.**

   This was a wrong formulation of the sentence. The opposite is the case. We rephrased to:
   *"Since then, several applications based on this method were developed and are routinely be used, e.g. for trace gas measurements (Abrams et al., 1996; Wang et al., 1996; Clerbaux et al., 2003; Bourassa et al., 2005; Fu et al., 2007)."*

7. **Page 5, line 5. The use of the acronym SZA and the symbol _0 for the solar zenith angle is redundant. Remove the acronym.**

   The acronym was replaced by the symbol.

8. **Page 6, figure 2. In the lower part of the figure it will be more convenient to plot the relative differences normalized to the Radiance. That will help with the corresponding discussion in lines 13-16. Also, some text is missing in the figure caption.**

The reviewer is right. Using the relative instead of the absolute differences in radiance makes the difference more clear. We, therefore, changed the plot according to the reviewers suggestion. The caption was extended.

9. **Page 6, line 3. Replace "wavelengths less. " by "wavelengths lower…"**

Replaced.

10. **"The RTS suggest that off-nadir observations at near infrared wavelengths ($\lambda$ > 900 nm) are more suitable for the detection of SVC and cirrus."**

Sentence is replaced by the reviewers comment.

11. **Page 8, figure 4 and lines 9-13. Because of the different values of I under the different constraints you should consider providing the sensitivity in percentages.**

Each panel in Figure 4 was calculated for a cirrus of fixed optical thickness. Therefore, using percentages instead of absolute values would not change the presentation and only scale the values. As the plots also aim to compare the four independent cases of different $\tau$ and $\Theta_0$, we prefer to stick with the absolute units in order to allow such a comparison. A normalization of the individual cases would remove this information. However, to improve the readability of the plot, we changed the illustration to 1d plots instead of the original color-coded 2d plots. This will make a comparison of the values between the panels easier.

12. **Page 9, line 2. Do you mean "thick clouds, for larger optical thickness…" here?**

Sentence has been changed.
*"While sideward viewing measurements are predicted to become saturated for thick clouds, for low tau the optimal $\Theta_V$ is about $\Theta_V = 60°$ with the largest $\varepsilon_\tau$ occurring for $\varphi$ between 0° and 60°."*

13. **Page 9, line 13. Remove "especially"**

Removed.

14. **Page 9, line 25. You should consider include a plot with the steepest derivative (maybe a subplot in Figure 5?)**

We are not sure, what the reviewer exactly meant by this comment. Figure 5 show the linear increase of the measured upward radiance caused by an increase of the surface albedo. In all cases, the increase is almost linear and, therefore, no steepest derivative exists. Only for each case one derivative can be calculated and is given in Table 1.

**15. Page 11, figure 6. Please, include a subplot with the relative differences between the different ice crystals. This will help with the discussion in lines 8-13.**

We agree, that relative differences will enhance the illustration of the differences between simulations with different ice crystal shapes and added such as subplot.

**16. Page 12, line 6. "were investigated"**

Changed.
*"…mid-latitudes were investigated in March and April…"*

**17. Page 12, line 11. Provide references for SMART and the calibration procedure.**

The SMART instrument characteristics and the calibration procedure are given in Section 3.1. So we think there is no need to give additional reference about the calibration here. Here we only added a reference introducing SMART in general.
Wendisch, M., Müller, D., Schell, D., and Heintzenberg, J.: An airborne spectral albedometer with active horizontal stabilization, J. Atmos. Oceanic Technol., 18, 1856–1866, 2001

**18. Page 12, line13-14. Provide references for the mini-DOAS and the DOAS technique.**

The DOAS instrument characteristics and the DOAS technique are discussed in Section 3.2. Here we only added a reference introducing the mini-DOAS in general.
Hüneke, T.: The scaling method applied to HALO measurements: Inferring absolute trace gas concentrations from airborne limb spectroscopy under all sky conditions, Ph.D. thesis, Ruperto-Carola University of Heidelberg, Germany, 2016.

**19. Page 13, line 21. The symbol ILmD has not been defined before. Please, define.**

Thanks for finding this shortcoming. ILmD is the upward radiance measured by the mini-DOAS in off-nadir direction. We rephrased to:
*"…applies least square retrievals on the spectral shape of the observed upward radiance ILmD by the mini-DOAS in sideward orientation…"*

**20. Page 14, line 26. Why are multiple scattering effects neglected?**

Multiple scattering effects are not neglected. This impression might have come up due to the unclear wording. As the apparent transmission is not needed to understand the lidar method, we deleted this statement.

*"Best compensation of the multiple scattering decay below the cloud is found for $r_{eff}$ = 35+-µm in good agreement with the climatological values proposed by Bozzo_2008. The mean correction factor for the data set shown in this paper was 7%."*

**21. Page 16, Figure 8. Can you add the error bars to the plots? Especially to plots b and d. Idem for figure 9.**

The uncertainty range has been added by shaded areas in both figures.

**22. Page 19, lines 26-27. Please, elaborate the statement "These stop criteria determine the accuracy of the iterative retrieval."**

We tried to rephrase the statement and added explanations to illustrate the iteration process better.

*"The iteration of tau is repeated until the change of $\tau_n$ between two iteration steps is smaller than 5% or a limit of n>100 iteration steps is reached. These stop criteria determine the accuracy of the iterative retrieval. If a lower relative stop criteria (change of $\tau_n$ smaller than 5% between two iteration steps or more then 100 iteration steps) is used the iteration may come closer to the true searched value and the retrieval accuracy increases as well as the necessary iteration steps and the computational time. To limit the computational time, the second stop criteria is used to limit the maximum number of iteration steps."*

**23. Page 20, lines 1-15. What happens for off-nadir observations?**

Right, we missed to add the same analysis for the retrieval using the sideward observations. In the revised version these information are added and emphasize the benefits of the sideward measurements:

*"Simulations show, that for $\tau$ = 0.5 the difference of $I^N_{RTS,1600}$ in nadir direction is only 0.1 mW when changing r_eff from 10 µm to 20 µm indicating the low sensitivity of r_eff retrievals at this wavelength. Therefore, a reliable retrieval of r_eff with reasonable accuracy is not feasible. For $I^V_{RTS,1600}$ the difference is 1.4 mW m$^{-2}$ sr$^{-1}$ and about a magnitude larger indicating that a retrieval of r_eff might be reasonable. However, in order to be consistent between both nadir and sideward viewing retrieval, r_eff has been fixed. A value of r_eff = 30 µm was chosen, a typical value of ice crystals observed by in-situ measurements during ML-CIRRUS Voigt et al., 2016. Therefore, the influence of an invalid assumption of r_eff on the iterative retrieval is analyzed. For this purpose the retrieval is tested for a typical cirrus of τ=0.3 and is run with three different assumptions of r_eff of 20 µm, 30 µm, 40 µm, representing the uncertainty of r_eff. These simulations imply that the retrieved tau changes only by +-0.02 between smallest and largest r_eff, resulting in a relative error in τ of 6.7%. The uncertainty in measured $I^N_{RTS,1600}$ and $I^V_{md,1600}$ causes a retrieval uncertainty of less than τ= +- 0.2. This justifies the fixed choice of r_eff in this specific cloud case."*

**24. Page 21, Figure 12. Axis labels are missing.**

Labels are added to the plot.

**25. Page 22, lines 14-15. Are these average values obtained for the coincident measurements only? Otherwise, comparing the different values is not realistic. Especially for the DOAS off-nadir, which have a smaller temporal resolution and does not capture all the variability observed during the analyzed period.**

The reviewer is right and the method to calculate the averages is now included in the manuscript.
*"Average τ are calculated for the filtered time period (indicated by the grey box in Fig. 14 for each instrument. Due to different sampling intervals, a different resolution and number of observations are included in the averaging calculations."*

**26. Page 23, line 3. A more in-depth analysis of the uncertainty will be useful, mainly for inter-comparison purposes between the different datasets presented in figure 14.**

More detailed explanation of the error estimation is added.
"The uncertainty range of tau is determined by running the retrieval twice with a bias of measured $I^N_{S,1180}$ with +-14.5% uncertainty at 1180 nm wavelength as upper and lower border. The resulting upper and lower retrieved tau represent the retrieval uncertainty."

**27. Page 23, line 9 and figure 14. It looks like there is a better agreement between the DOAS off-nadir and the reference WALES than between the DOAS off-nadir and DOAS nadir or SMART. Can you comment something on that? Can you further discuss the advantages and disadvantages of having nadir and off-nadir measurements and link it with the sensitivity analysis in section 2?**

Resulting from the different observation geometries of the nadir looking sensors and the mini-DOAS sideward sensors different cloud scenes are probed. This can lead to the different values of retrieved tau and the good agreement between mini-DOAS sideward and WALES. Please have a look to the added description.
*"Average τ are calculated for the filtered time period (indicated by the grey box in Fig. 14 for each instrument. Due to different sampling intervals, a different resolution and number of observations are included in the averaging calculations. The retrieved average of τ at 532 nm is 0.54+-0.2 (SMART), 0.49+-0.2 (mini-DOAS nadir spectrometer), 0.27+-0.2 (mini-DOAS sideward viewing spectrometer) and 0.32+-0.02 (WALES). The results indicate a reasonable agreement of τ retrieved by SMART and mini-DOAS nadir channel, while lower τ are inferred from mini-DOAS sideward viewing and WALES measurements. Taking the WALES measurements as a reference, the measurements of SMART and mini-DOAS overestimate τ. However, by estimating the uncertainty of the mini-DOAS and SMART basing on RTS, the measurement error of $I^N_{S,1180}$ (14.5%) by SMART results in an uncertainty range of retrieved τof +-0.2, which covers the values of τobtained by WALES. The uncertainty range of τis determined by running the retrieval twice with a bias of measured $I^N_{S,1180}$ with +-14.5%*

*uncertainty at 1180 nm wavelength as upper and lower border. The resulting upper and lower retrieved τ represent the retrieval uncertainty. The mean τ inferred from the mini-DOAS sideward viewing observations is significantly lower than measured by SMART and mini-DOAS nadir measurements. Differences in τ range up to +-0.73 between SMART and mini-DOAS sideward viewing observations. This may result from the different FOV of the sideward viewing geometry that does not observe the exact same clouds as SMART and nadir channels did. With the scanning sensors orientated to starboard the sideward viewing retrieval corresponds to cirrus 8 km east of the flight track. As the MODIS satellite image in Fig. 12 indicates, the cirrus becomes slightly thinner towards east, which possibly is due to the lower values of τ. Other potential reasons are the assumed ice crystal shapes for the RTS and different field-of-view of the passive and active remote sensing instruments. On the other hand, the agreement between mini-DOAS sideward observations and WALES is significantly better. The maximum difference of τ between mini-DOAS sideward channels and WALES is +-0.25 while the difference between the mean values is +-0.05 (15.6%). With WALES and mini-DOAS measuring in different viewing geometries but showing better agreement, the differences of τ retrieved by SMART is most likely caused by uncertainties in α. As discussed in Section 2.3, nadir observations are stronger affected by α than sideward observations. This is confirmed by the smaller differences between WALES and mini-DOAS sideward observations and indicates the advantage of the sideward viewing retrieval due to a reduced surface influence and lower retrieval uncertainty."*

28. **Page 23, lines 20-21. This statement is not clear. If the data points contaminated by the second cloud layer are excluded from the calculations, what do you mean here?**

This statement was misleading. The section of the time series used to calculate the average values was carefully selected for an area where no second cloud layer was observed. This selection bases on the analysis of the WALES profiles. However, due to the larger FOV of the passive sensors, there is the chance that SMART and mini-DOAS are still contaminated by such a second cloud layer but not WALES. We extended the description of the data selection for the calculation of the averages.
All points which differed clearly were excluded from the calculations. Nevertheless there is a slight chance that few points were classified as cirrus but actually belong to the second cloud layer. This is mostly due to the fact that they could not be separated definitely and because the SMART and mini-DOAS sensors have a larger FOV compared to WALES.
*"...These data points are excluded from the following analysis. Nevertheless there is a slight chance that few points were classified as cirrus but actually belong to the second cloud layer. This is mostly due to the fact that they could not be separated definitely and because the SMART and mini-DOAS sensors have a larger FOV compared to WALES."*

29. **Page 24, lines 10-12. This is not clear either. From the results and the discussion presented before, it looked like you were using the wavelength of 532 nm for all the instruments. Please, clarify where necessary.**

The reviewer is right, this statement might be misleading. The measurements of the different sensors have been analyzed at different wavelengths (1180 nm for SMART and

mini-DOAS and 532 nm for WALES). However, the retrieved cirrus optical thickness always refers to 532 nm. Therefore, the retrieval for the passive remote sensing of SMART and mini-DOAS consider simulations at both wavelengths. In the radiative transfer simulations the cirrus optical thickness is defined and changed at 532 nm while the simulations and measurements at 1180 nm were compared to find the correct solution:

 *"Additionally, the different wavelengths of the measurements may introduce biases in the retrieved tau due to different penetration depth of the reflected radiation into the cloud (Platnick, 2000). Therefore, the wavelength selection defines the layer in the cloud which is probed. While WALES uses backscatter measurements at lambda= 532 nm and lambda= 1064 nm the measurements of $I_{S,1180}$ by SMART and mini-DOAS are performed at $\tau$= 1180 nm. Although the retrieval accounts for the wavelength dependence of scattering, absorption and refraction on ice crystals (Takano and Liou, 1989; Yang et al., 2013) by scaling the retrieved tau at $\lambda$ = 1180 nm to $\lambda$= 532 nm to make it comparable between the instruments."*
*"The retrieval of tau by SMART and mini-DOAS bases on the measurements at $\lambda$ = 1180 nm and is scaled to $\lambda$ = 532 nm to consider the wavelength dependence of tau and to be able to compare it with WALES measurement at $\lambda$ = 532 nm. Therefore, the retrieval considers RTS at both wavelengths. In the RTS $\tau$ is defined and changed at $\lambda$ = 532 nm while the measurements are compared to simulations at $\lambda$ = 1180 nm to determine the correct solution."*

30. **Page 26, line 14. Agreement is within the uncertainty but I would not consider a 66.6Numerical values for the differences between DOAS nadir and Wales and DOAS off-nadir should be included separately. Relevance was given to the comparison between the nadir and off- nadir observations in the sensitivity analysis and it will be interesting to do a clear distinction also for the in-situ airborne data and include a significant conclusion at this respect.**

To add an explicit comparison of the DOAS sideward and nadir results we added an additional 1:1 plot in Figure 15. Here the calculated mean optical thickness values have to be analyzed as done in section 5.2.1. Alternatively, we show a comparison between DOAS-sideward and WALES measurements in the additional 1:1 plot.  A good agreement was found indicating also that DOAS-sideward and DOAS-nadir will have an agreement similar to the comparison of WALES and DOAS-nadir. This illustrates the capability of sideward measurements to observe optically thin cirrus and the higher accuracy of this method for optical thin clouds. There conclusions have been added.

---

## Author Comment (AC2) · 24 Feb 2017

We thank the reviewer for the encouraging words and fort he helpful comments which improved the manuscript noticeably. By adding some more explanations and hints from a person not involved in the manuscript preparation enhanced the understanding for the reader.

The replies of the reviewer comments are given in the following manner: Reviewer comments are printed in bold, are labeled, and are listed in the beginning of each answer. The reviewer comments are followed by the author comments and revised parts of the paper. The revised parts of the paper are written in quotation marks and italic letters.

**Comments:**

1. **Page 2: there are significant discussions for SVC, but techniques presented here a not suitable for dealing with cirrus with such small optical depth- large uncertainties among them.**
   The reviewer is right. In the manuscript a case study of an observed cirrus with higher optical thickness than SVC is presented. However, the sensitivity study (Figure 3) shows that the sensitivity of measured radiance is higher for sideward measurements compared to nadir measurements for cirrus with low optical thickness up to $\tau=1$. The observed cirrus case showed $\tau$ in the range of 0.2-1.0 and, therefore, is suitable to test the different observation geometries although it is not in the range of SVC. Using a cirrus with higher optical thickness than SVC as a first test case has the advantage, that the measurement uncertainties are less important for the retrieved $\tau$ (higher reflected radiance way above the instrument noise level). Therefore, we think that using a moderate thick cirrus is most suited here. Additionally, Sections 4 and 5 (especially 5.2.1) include a more detailed investigation of the mini-DOAS sideward observations, indicating and emphasizing the advantages of these observations compared to measurements in nadir geometry for all types of thin cirrus clouds including SVC. Furthermore two plots were added to Figure 15 to show the agreement of tau between WALES and the mini-DOAS sideward measurements.

   *"The retrieval using mini-DOAS sideward channels is also successful demonstrated for a reduced set of observations limited to $\Theta_V$ between 85° and 90°. Differences in $\tau$ range up to +-0.73 between SMART and mini-DOAS sideward viewing observations and are partly caused by the different viewing geometries. First, the sideward telescopes view into starboard direction, probing the cirrus cloud top at approximately 8000 m aside the flight track. Second, the nadir observations may suffer from uncertainties in $\alpha$ while the sideward observations are less effected by changes in $\alpha$. Even for sea surfaces as presented here, $\alpha$ may change due to different wind speeds. Other potential reasons are the assumed ice crystal shapes in the RTS and different field-of-view of the passive and active remote sensing instruments. This conclusion is apparent from different probability distributions. While SMART and mini-DOAS show a median around $\tau=0.4$, the median for WALES is shifted to lower $\tau$ around 0.2, indicating that WALES observed small $\tau$ more frequently. The difference of mean values of $\tau$ between mini-DOAS sideward channels and WALES is smaller with +-0.05 (15.6%). This shows the advantage of the sideward viewing retrieval due to a reduced surface influence and lower retrieval uncertainty, because of high $\varepsilon_\tau$ compared to the nadir measurements."*

2. **Page 2, last sentenceăˇAˇTit is not an accurate statement if you consider passive sensor measurements.**
Sentence is rephrased and the word "inherently" is removed.
*"While satellite observations are suited to study the global coverage of cirrus, their spatial and temporal resolution is still limited and can not resolve the high spatial variability of cirrus."*

3. **Page 3: Lines 5-6: the cirrus optical thickness of water clouds does not make senseăˇAˇTre-write.**
The word "cirrus" is removed from the sentence as water clouds can not be cirrus.
*"For nadir measurements tau and the effective radius r_eff of liquid water droplets can be retrieved by the bi-spectral reflectivity method after Twomey, 1980 and Nakajima, 1990. Ou et al., 1993, Rolland et al. 2000, and King et al., 2004 adapted this method for ice clouds by introducing some modifications with regard to the thermodynamic phase and crystal shape of the ice particles."*

4. **Page 4: Line 18: If you conclude that it is impossible here. You don't need any further study in this paper. Yes, it is challenging, which indicates that we need more observational constrains to improve the retrieval.**
This statement did not ment that cirrus retrieval are in general impossible. "Impossible" referred to the worst conditions where uncertainties may get too large. Most observations will provide conditions where a retrieval is possible but with uncertainties. Therefore, the conclusion was rewritten:
*"In a worst scenario, all these effects render retrievals of τ to become rather inaccurate. However, observations in sideward or limb viewing direction and improvements of retrieval techniques may overcome these limitations."*

5. **Page 4, Lines 19-21: The statements here are not accurate. Off-nadir measurements are widely used for space-base cirrus remote sensing. As you know, most satellite passive sensors are wide swath measurements .**
That is correct. Sideward can mean 1° off-nadir viewing direction. With "sideward" we address measurements close to limb direction (90° viewing direction). We refrain of using the term "limb" measurements because the measurements presented in the manuscript had been performed at viewing angles less than 90°, not limb. We now included an explanation on what we define as sideward measurements and in case literature is discussed where real limb-measurements are applied, we now kept writing limb viewing angle. The word "off-nadir" was replaced by "sideward" in the entire manuscript to avoid misunderstanding.
*"Limb measurements of SVC and cirrus were first introduced and utilized for satellite measurements by Woodbury, 1986. Since then, several applications based on this method were developed and are routinely be used, e.g. for trace gas measurements (Abrams_1996, Wang_1996, Clerbaux_2003, Bourassa_2005, Fu_2007).*
*Many trace gas retrievals from aircraft, balloons and satellites are based on ultraviolet (UV)/ visible (VIS)/ near infrared (IR) sideward viewing measurements in combination with*

*differential optical absorption spectroscopy (DOAS), e.g. performed by Platt_2008. Compared to nadir observations, radiance measurements in limb or sideward viewing geometry are supposed to be more sensitive to optical thin clouds due to their observation geometry. One recent study was accomplished by Wiensz 2013 who used satellite limb measurements especially for SVC investigation in the tropical tropopause layer. This data source improved SVC observations with respect to cloud climatology and microphysics."*

6. **Page 4, Line 25, "highly sensitive": An overstatement. Yes, it is more sensitive, but it is highly dependent on the magnitude of off-angle.**
Replaced the word "highly" by "more" to avoid the overstatement. The influence of the observation angle is shown later in the different simulations of the sensitivity study.
*"Compared to nadir observations, radiance measurements in limb or sideward geometry are supposed to be more sensitive to optical thin clouds due to their observation geometry."*

7. **Page 5, Line 15: What does "F" in "FDISORT" mean?**
FDISORT is the Fortran 77 version of the original DISORT solver:
*"The Fortran 77 discrete ordinate radiative transfer solver version 2.0 (FDISORT 2) after Stamnes 2000 is chosen."*

8. **Page 6: Figure 2 caption in the PDF misses words.**
Caption is corrected.

9. **Page 6, line 15: The statement of "cirrus can not be detected" is not accurate. Cirrus is a general category including high clouds with optical depth up to 3.**
Sentence is formulated more precisely just refereeing to sub visible cirrus:
*"Therefore, at λ= 532 nm SVC with τ= 0.03 which is presented in the simulations can not be detected."*

10. **Page 7, lines 4-5: To draw this conclusion, you'd better to present calculation results with a higher optical depth.**
Thanks for this suggestion. We added simulations for a cirrus with τ=2.0 in the revised manuscript what helped to illustrate the differences to the SVC case. The simulations for τ=2.0 show that the effect of Rayleigh scattering is significant reduced at 532 nm and a separation between cloudy and clear-sky is possible for such clouds. Nevertheless, the relative difference between cloudy and clear-sky case is still more pronounced for the radiance at 1180 nm and emphasizes the conclusion:
*"For comparison, simulations of a thicker cirrus with τ= 2.0 are presented in Figure 2 (b). Here, the influence of the Rayleigh scattering at λ= 532 nm is reduced and a distinction between cloudy and clear-sky conditions becomes possible. However, the relative difference between cloudy and clear-sky is still more pronounced at λ= 1180 nm.*
*The RTS suggest that sideward viewing observations at near IR wavelengths (λ> 900 nm) are more suitable for the detection of SVC and cirrus. As a result the retrieval in Section 4 is performed at 1180 nm and 1600 nm wavelength in the IR region which are sensitive to τand r_eff and not disturbed by Rayleigh scattering."*

11. **Page 7, line 15: This statement does not consistent with the statements in the next paragraph.**
Statement was rephrased and specified for cirrus with optical thickness below 1. It was added that for clouds with optical thickness larger than 1 the sensitivity of sideward observations is in the same range compared to nadir measurements. This shows that sideward measurements are most suited and applicable for cirrus with tau below 1.
*"Due to the apparent longer LOS for both $\Theta_0$, sideward viewing sensor orientations yield larger $\varepsilon_\tau$ of simulated $I^V_{RTS}$ as compared to the nadir geometry for cirrus clouds with $\tau<1$ which includes SVC. This indicates that sideward measurements are most suited to retrieve tau below 1 and for the detection of SVC. The almost linear increase of the nadir radiance $I^N_{RTS}$ indicates a constant $\varepsilon_\tau$ tau for the investigated range of tau and $\Theta_0$. For $\tau >= 1$ the sensitivity of sideward viewing observations is in the same range compared to nadir measurements or slightly lower depending on the combination of $\Theta_0$ and $\Theta_V$."*

12. **Page 8, line 1-2: To draw this conclusion, you need to make many assumptions.**
Conclusion is extended and explained in more detail.
*"For low tau and a high sun, the highest $\varepsilon_\tau$ is given for the sideward viewing geometry ($\Theta_V = 78°$) for $\tau <= 1$. A similar pattern emerges for low Sun ($\Theta_0 = 75°$) resulting in larger $\varepsilon_\tau$ and a steep decrease for increasing $\tau$. It shows that $\varepsilon_\tau$ decreases with tau and for $\tau < 2$ drops below $\varepsilon_\tau$ of nadir measurements. The sensitivity of I with respect to $\tau$ can also be interpreted in terms of the uncertainty of retrieved $\tau$ related to an initial uncertainty in measured I. The higher $\varepsilon$ the weaker the impact of uncertainties in the measurements on the uncertainties of the retrieved $\tau$. As shown in Fig.3 (b), a high $\varepsilon_\tau$ is calculated for $I_{RTS,1180}$ for $\tau <= 1$ and indicates a lower measurement uncertainty. Therefore, sideward viewing observations at $\lambda= 1180$ nm allow a more accurate determination of $\tau$ compared to nadir observations for optical thin clouds with $\tau <= 1$."*

13. **Page 8, line 7: Based on the statement, it seems that you don't consider angle smaller than 60 degree as the off-nadir observations. That is not right.**

See reply to comment 5.
Sideward measurements with angles smaller than 60 degree are not that sensitive as compared to larger angles. This does not necessarily mean that they are not considered but they are unfavorable. The plot should show that depending in the optical thickness and the relative solar azimuth angle the best viewing angle should be selected to reach the highest sensitivity which results in the lowest relative measurement errors and better retrieval results.
*"For $\tau= 0.1$ and $\Theta_0= 25°\$$ (Fig. 4 a), $\varepsilon_\tau$ ranges between 5 and 66 mW $m^{-2}$ $nm^{-1}$ $sr^{-1}$. For larger $\Theta_V$ (sideward viewing observations) $\varepsilon_\tau$ increases significantly reaching the maximum for $\Theta_V =90°$ and $\varphi = 0°$. Observations under these angles are better suited in comparison to other angle combinations as they enable to achieve the largest possible $\varepsilon_\tau$ and reduced relative measurement errors which results in increased retrieval accuracy."*

14. **Page 9, line 9-10: It is hard to understand this sentence.**

In the revised manuscript we rephrased the sentence:

*"Measurements in sideward viewing geometry strongly dependent on $\Theta_V$ especially around $\Theta_V = 90°$. In order to avoid spurious results by mispointing with the sensor, a careful alignment of the optical sensor and an accurate determination is required. Considering these findings, the retrieval of tau in Section 4 is performed for $\Theta_V <= 60°$ only."*

**15. Page 12, figure 7: It is hard to see the location of the optical port in (b). A better figure may be needed.**
Location of the ports is highlighted in an updated figure and is hopefully visible now.

**16. Page 12, line 12: UV and VIS were defined early.**
Removed.

**17. Page 12, line 13: DOAS was defined early. –Avoid multiple definitions.**
Removed.

**18. Page 15, line 3: "cross-calibrate both instrument" is no right. As you discussed in the paper, SMART is lab calibrated.**
The sentence was rephrased:
*"Since no radiometric calibration is available for mini-DOAS, simultaneous measurements of SMART and mini-DOAS are used to cross-calibrate the mini-DOAS with SMART."*

**19. Page 16, lines 1-2: Giving absolute numbers are needed, but it will be good to present relative differences too.**
We added relative differences for the nadir and sideward cross-calibration in the revised text.
*"…, which results in relative differences of 5.4% at λ= 1180 nm and 1.9% at λ= 1600 nm compared to the SMART absolute values."*

**20. Page 18, line 3: Based on Fig. 10, I'd like to say that 2.9 is a big number, which is difficult to support the stable calibration consistent.**
Yes, 2.9 actually is a big number considering the aim to retrieve cirrus optical thickness with reasonable accuracy. We, therefore, included estimates of uncertainties in the retrieval of tau that would be caused by such uncertainties in the calibration.
However, the comparison of the calibrations did not suggest to use a calibration that was done long before or after the measurements. This was now emphasized in the revised manuscript.
Nevertheless, considering the original purpose of the mini-DOAS to remain a precise wavelength calibration for DOAS observations but without need to relying on a radiometric calibration because relative measurements are analyzed, the relative good stability of the calibration was surprising. The radiometric calibration can change between campaigns due to instrument removal and modification but also between flights by switching the instrument on and off. As the deviation is 2.9 and ranging in the uncertainty range of SMART the stability is good taking SMART as a reference. If no subsequent calibration of the mini-

DOAS would be available, radiometric measurements would still be possible considering the uncertainty of 2.9 mW, what may in some application be sufficient.

**21. Page 19, Line 26: The statement here is not consistent with the lowest box in Fig. 11.**
Illustration is corrected so it is in agreement with the text.

**22. Page 20, line 20: Even for lidar guy, it is hard to see contrails in Fig. 13. How about to plot Fig. 13 as a color figure to make the fine feature easy to identify.**
B/W-plot was replaced by a color plot.

**23. Page 22, line 22: For cirrus cloud optical depth around 1, it is hard to claim that the lower layer is obscured by the upper cloud layer. The lower layer can be clearly identified from lidar image.**
This is correct. We did not clearly separate this discussion between lidar and passive sensors. Of course, the lidar can provide vertically resolved measurements and, therefore, is able to separate the second cloud layer. This means that WALES can determine tau of the cirrus (without the lower layer) correctly. On the other hand SMART provides only vertically integrated information as it measures the sum of reflected radiation from both clouds. This partly explains the bias in the retrieval. The paragraph was rephrased.
*"A second segment with higher retrieved τ is likely due to an underlying cirrus between 8.5 km and 9.5 km altitude that is also obscured to the detection by WALES. Therefore, a positive systematic offset of the retrieved τ occurs for SMART and mini-DOAS. These data points are excluded from the following analysis. Nevertheless, there is a slight chance that a few cloud fragments of these second cloud layers are still affecting the SMART- and mini-DOAS retrieval. Both passive sensors have a larger FOV compared to WALES and, therefore, are more likely sensitive to cloud layers located below the cirrus."*

**24. Page 23, line 3: Is 10% here mean error or random error? You need to explain the +0.2 overestimation.**
The measurement uncertainty of SMART is 14.5% and not 10%. This was corrected in the manuscript. The uncertainty of ±0.2 results from the uncertainty in the measured upward radiance of 14.5%. For this estimation the retrieval was performed twice with a bias of *I* with ±14.5% uncertainty as upper and lower border. We rephrased the paragraph to make this procedure more clear:
*"The uncertainty range of tau is determined by running the retrieval twice with a bias of measured $I^N_{S,1180}$ with +-14.5% uncertainty at 1180 nm wavelength as upper and lower border. The resulting upper and lower retrieved tau represent the retrieval uncertainty."*

**25. Page 23, lines 16-23: Which kind of calibration errors explain the good linear correlations and 0.66 or 0.69 slopes?**
SMART and mini-DOAS relay on a large field of view, in the range of several tenth of meters depending on the distance between sensor ans cloud top, compared to WALES which has a narrow opening angle of 0.08°. Additionally, SMART and mini-DOAS are passive remote

sensing instrument measureing the scattered radiation from the sun which is effected by the entire atmosphere. Contrarily, the WALES measurement is influenced by interactions in the smaller field of view only. Also the food print at cloud top is much smaller from WALES compared to SMART and mini-DOAS. Therefore, WALES has a higher horizontal resolution. This becomes visible in the propability density functions of the three instruments, where the median for WALES is shifted to lower optical thickness, indicating that WALES measured more values of low tau and even cloud free regions. On the other hand SMART and mini-DOAS measurements average over larger areas and do not represent these small fluctuations which can explain the linear offset.

Due to the larger FOV of the passive sensors, there is the chance that SMART and mini-DOAS are still contaminated by a second cloud layer but not WALES. We extended the description of the data selection for the calculation of the averages.

All points which differed clearly were excluded from the calculations. Nevertheless there is a slight chance that few points were classified as cirrus but actually belong to the second cloud layer. This is mostly due to the fact that they could not be separated definitely and because the SMART and mini-DOAS sensors have a larger FOV compared to WALES.

26. **Page 24, lines 10-11: For large ice crystals, why do you expect optical depth difference between 532 nm and 1180 nm?**

For the retrieval of optical thickness the wavelength applied in the retrieval has to be considered as the ice crystal extinction is wavelength dependent (Takano and Liou, 1989, Yang et al., 2012).

Although, this dependence was considered by scaling all results to lambda= 532 nm, the reflected radiance at different wavelengths used for the retrieval have different vertical weighting functions (Platnick, 2000). Depending on the wavelength, the penetration depth of solar radiation into a cloud can vary. While wavelengths close to the UV have a higher penetration depth compared to wavelength close to the infrared region.

This effect might be small in case of vertically homogeneous cirrus but is a potential uncertainty source which have to be considered for vertical inhomogeneous cirrus observed here.

*"Additionally, the different wavelengths of the measurements may introduce biases in the retrieved tau due to different penetration depth of the reflected radiation into the cloud (Platnick, 2000). Therefore, the wavelength selection defines the layer in the cloud which is probed. While WALES uses backscatter measurements at λ= 532 nm and λ= 1064 nm the measurements of $I_{s,1180}$ by SMART and mini-DOAS are performed at λ= 1180 nm. Although the retrieval accounts for the wavelength dependence of scattering, absorption and refraction on ice crystals (Takano_1989,Yang_2013) by scaling the retrieved τ at λ= 1180 nm to λ = 532 nm to make it comparable between the different instruments."*

*Takano, Y. and K. Liou, 1989: Solar Radiative Transfer in Cirrus Clouds. Part I: Single-Scattering and Optical Properties of Hexagonal Ice Crystals. J. Atmos. Sci., 46, 3–19, doi: 10.1175/1520-0469(1989)046<0003:SRTICC>2.0.CO;2.*

*Yang, P., L. Bi, B. Baum, K. Liou, G. Kattawar, M. Mishchenko, and B. Cole, 2013: Spectrally Consistent Scattering, Absorption, and Polarization Properties of Atmospheric Ice Crystals at Wavelengths from 0.2 to 100 μm. J. Atmos. Sci., 70, 330–347, doi: 10.1175/JAS-D-12-039.1.*

*Platnick, S., 2000: Vertical photon transport in cloud remote sensing problems. J. Geophys. Res. Atmos., 105, 22919-22935, doi: 10.1029/2000JD900333.*